# Understanding Dimensional Collapse in Cross-Modal Feature Distillation

## Abstract

To overcome limited computing resources and the complexity of sensor configurations in deploying multi-modal neural networks in real-world applications, *cross-modal knowledge distillation* (CMKD) aims to transfer valuable information from a pretrained teacher model to a deployable student model with the target modality. Despite the successful applications of CMKD in various fields, our understanding of knowledge transfer across different modalities remains insufficient to fully explain the efficacy of feature distillation. In this work, we investigate the relationship between the distributional shifts across modalities, referred to as the *modality gap*, and its impact on the effectiveness of CMKD, particularly focusing on the problem of *cross-modal feature distillation*. We first hypothesize and empirically validate that the modality gap between the teacher and student causes *dimensional collapse* in the student's feature space. To prevent such inefficiency, we propose a *Cross-modal Information Bottleneck Approximation* (CIBA) scheme aimed at extracting and transferring modality-general features from the teacher model. Lastly, we experimentally demonstrate that our distillation strategy effectively reduces the dimensional collapse in the student model, thereby achieving improved performance for various real-world multi-modal datasets.

## 1 Introduction

Multi-modal learning aims to extract comprehensive features from multiple sensory inputs (Huang et al., 2021; Ngiam et al., 2011) and has demonstrated its effectiveness in fusing data from various domains, including images, audio, texts, and 3D point clouds (Anderson et al., 2018; Driess et al., 2023; Li et al., 2022d; Livingstone & Russo, 2018; Xue et al., 2021). However, integrating multi-modal inputs inevitably increases the complexity of models and inference time during deployment in real-world applications.

As a remedy, prior works leverage *cross-modal knowledge distillation* (CMKD) to transfer valuable information from a pretrained teacher model to a student model by forcing the student to mimic the teacher's behavior, and only deploy the student model with the target modality (Hong et al., 2022; Ren et al., 2021; Thoker & Gall, 2019). While CMKD has shown practical value in various applications, the efficacy of knowledge transfer and its internal mechanisms remain inadequately explored (Gou et al., 2021). One recent study (Xue et al., 2022) introduced the concept of modality-general and specific features, highlighting the proportion of general features as decisive factors for the quality of CMKD. However, we still lack a clear explanation of why and how these factors affect the efficacy of CMKD, as well as the root causes of such ineffectiveness.

In this work, we present an in-depth analysis on how distributional shifts across different modalities, referred to as *modality gap*, leads to *dimensional collapse* (Hua et al., 2021) in the student model and results in suboptimal knowledge distillation performance. In particular, we focus on *cross-Modal feature distillation* (CMFD), where dimensional collapse can significantly deteriorate the quality of the distillation results. Let us assume that the features of the teacher model include both modality-general and modality-specific knowledge (Xue et al., 2022) as depicted in Fig.1-(a). Applying typical feature distillation strategies (*e.g.*, mean-squared error (MSE) loss, cross-entropy (CE) loss) leads to biasing of the student's feature space to the modality-general features (*e.g.*, green area in Fig.1), which are the only transferable knowledge from the teacher. Thus, the features of the student model only span sub-dimensions and lead to dimensional collapse as exemplified in Fig.1-(c).

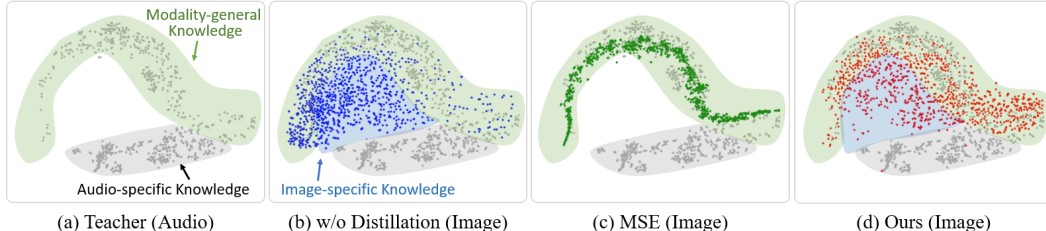

(a) Teacher (Audio)     (b) w/o Distillation (Image)     (c) MSE (Image)     (d) Ours (Image)

Figure 1: Distribution of features trained with and without distillation, for audio to image distillation on RAVDESS (Livingstone & Russo, 2018) dataset. During the distillation process, **(a)** teacher (audio) model was frozen and the distribution of teacher's features is plotted with gray dots in all sub-figures for comparison. Feature distributions from audio **(a)** and image **(b)** models exhibit overlap in some regions (green, modality-general) while others are modality-specific (gray and blue, modality-specific). A typical feature distillation **(c)** reduces the distributional diversity (*i.e.*, dimensional collapse) of the learned student features by biasing the student toward only general knowledge (green area). In this paper, we propose the Cross-modal Information Bottleneck Approximation (CIBA) scheme that effectively addresses dimensional collapse, as shown in **(d)**.

To prevent such inefficacy, we employ the *information bottleneck* scheme that extracts modality-general features and removes intractable modality-specific features (*e.g.*, gray area in Fig.1) from the teacher, then distill the shareable knowledge (*e.g.*, green areas in Fig.1) to only a sub-dimension of the student. This allows the student to effectively span both student-specific (*e.g.*, blue areas in Fig.1) and modality-general (*e.g.*, green areas in Fig.1) feature space, thereby enabling the learned features to cover broader regions of modality-general and specific knowledge areas more evenly. Please note the broader dispersion of feature embeddings depicted in Fig.1-(d) compared to Fig.1-(b) and (c). Our contributions can be summarized as follows:

- We theoretically and empirically investigate the impact of the modality gap on cross-modal knowledge distillation by examining dimensional collapse.
- We propose the **C**ross-modal **I**nformation **B**ottleneck **A**pproximation (**CIBA**), a novel knowledge distillation strategy that effectively extracts modality-general features from the teacher model and transfers them to sub-dimensions of the student's features.
- We validate our distillation approach on various real-world datasets, including RAVDESS (Audio-Image), MM-IMDB (Image-Text), nuScenes (LiDAR-Camera), VGG-Sound (Video-Audio).

## 2    RELATED WORKS

### 2.1    CROSS-MODAL KNOWLEDGE DISTILLATION

The primary goal of Cross-modal knowledge distillation (CMKD) is to enhance performance of the student model with the target modality by transferring valuable knowledge from the teacher's modality (Gupta et al., 2016). In light of its practical aspects, CMKD has been investigated for various applications such as multi-modal classification (Huo et al., 2024), video representation learning (Sarkar & Etemad, 2024), speech recognition (Jin et al., 2023), and emotional recognition (Zhang et al., 2022). More recently, CMKD has been extended to more challenging tasks such as 3D object detection (Chen et al., 2023; Wang et al., 2023; Li et al., 2022b) and 3D semantic segmentation (Sautier et al., 2022) based on multi-modal imaging sensors (*e.g.*, LiDAR, camera, radar).

While CMKD has achieved some success, it may lead to suboptimal distillation results, lacking adequate consideration of the distributional shifts across different modalities. A pioneering work (Xue et al., 2022) suggested that the success of CMKD largely depends on the extent to which modality-general decisive features are captured in the teacher network. Another recent work (Huo et al., 2024) empirically investigates that modality imbalance and soft label misalignment between the teacher and student modalities hinder output-level CMKD. However, we still lack a clear understanding of why and how such decisive features affect the efficacy of cross-modal *feature* distillation. In this work, we aim to address these questions by investigating the concept of the modality gap in relation to the dimensional collapse of learned student features. We further propose a novel distillation method to mitigate the effect of the modality gap and enhance CMFD performance.

## 2.2 DIMENSIONAL COLLAPSE IN FEATURE SPACE

Dimensional collapse happens when a model fails to fully utilize its capacity to encode information, leading to a reduction in the dimensionality of the learned feature space. (Hua et al., 2021; Jing et al., 2021; Li et al., 2022a). Several prior works have attempted to understand dimensional collapse. Jing et al. (2021) discover that strong data augmentation and implicit regularization of an over-parameterized model cause dimensional collapse in the self-supervised contrastive learning. To mitigate this, they propose DirectCLR, which directly optimizes the sub-dimensional representation vectors instead of fully utilizing an explicit trainable projector. Recent studies in self-supervised representation learning (Bardes et al., 2021; Zbontar et al., 2021) have explored methods to increase the expressiveness of learned features by applying regularization to maximize information. In this paper, we discover that the dimensional collapse also occurs in CMKD, and provide both theoretical and empirical analyses on this matter. Based on this observation, we suggest an information bottleneck approximation strategy to effectively alleviates the collapse issue.

## 3 DIMENSIONAL COLLAPSE IN CROSS-MODAL FEATURE DISTILLATION

### 3.1 PROPOSITION

We investigate the cross-modal feature distillation (**CMFD**) problem as the effect of dimensional collapsing can be decisive and clearly observed in the high-dimensional features than low-dimensional outputs. CMFD differs from traditional single-modal feature distillation methods (Heo et al., 2019), with the key distinction being that each of the teacher and student networks receives a different form of modality as input. (Hong et al., 2022; Li et al., 2022b; Xue et al., 2022). Although each modality is expected to be correlated with one another over the training data distribution, a certain level of knowledge contained in one modality (teacher) may not be transferable to the other modality (student). For example, a speaker's gestures can be observed in images, but not through audio. Recent studies have verified that the gap between the teacher and student modalities affects the efficacy of output-level CMKD (Xue et al., 2022). In this paper, it is argued that these claims can also be extended to feature-level CMKD, and the impact of the modality gap can be explained in relation to the dimensional collapse observed in learned student features.

**Claim 1.** *When a modality gap is present between teacher and student modalities, global feature distillation strategy[1] may result in the dimensional collapse of learned student features. Moreover, as the gap between modalities increases, dimensional collapse becomes more prominent.*

In the following section, we provide theoretical supports for our claim, and experimentally verify it using a synthetic dataset (Xue et al., 2022) where modality-general portion can be modified manually.

### 3.2 PROBLEM STATEMENTS

Suppose that datasets from student and teacher modalities are given by $X = [\mathbf{x}_1, ..., \mathbf{x}_N]$ and $X' = [\mathbf{x}'_1, ..., \mathbf{x}'_N] \in \mathbb{R}^{D \times N}$ respectively, where $N$ and $D$ denote the number of data samples and dimensionality of each sample, respectively. Each column of $X$ and $X'$ is paired each other. Student and teacher model have linear feature-extractor $W$ and $W' \in \mathbb{R}^{F \times D}$, respectively, where $F$ denote the dimensionality of feature.

In our general setting, we utilize the Mean Squared Error (MSE) loss for feature distillation (FD), as it is a widely employed function in the context of FD (Hafner et al., 2022; Hong et al., 2022; Lee et al., 2023). However, our claim can be extended to other global feature distillation losses, including cross-entropy, as evidenced in Sec.5 and Appendix E.5. MSE loss is typically defined as:

$$L_{FD} = \frac{1}{2} \sum_{i=1}^{N} \|W'\mathbf{x}'_i - W\mathbf{x}_i\|_2^2. \tag{1}$$

In order to solely examine the impact of FD on the student model, we did not consider task losses during theoretical derivation. However, we empirically demonstrate that our analysis can be extended to scenarios where task losses are applied, as demonstrated in extensive experiments in Sec.3.5 and 5.

---

[1]We refer to the prevalent distillation strategy that forces the features of the student model to mimic the whole features of the teacher model (*e.g.*, mean-squared-error, cross-entropy) as *a global feature distillation* strategy.

## 3.3 DERIVATION OF OPTIMAL STUDENT WEIGHTS

To analyze a student weight $W$ trained by Eq.1, we first derive the optimal solution $W^*$ for Eq.1.

**Lemma 1.** *If $X$ has full rank ($Rank(X) = D$), student weight $W$ is converged to $W^*$ such that*

$$W^* = W'X'X^T(XX^T)^{-1}. \tag{2}$$

**Lemma 2.** *Let $X$ be decomposed as $X = U\Lambda V^T$ by singular value decomposition. And define right inverse matrix of $\Lambda$ as $\Lambda^{-1}$ such that $\Lambda\Lambda^{-1} = \boldsymbol{I}_{D\times D}$. Then $W^*$ can be developed as*

$$W^* = W'(X'V\Lambda^{-1})(XV\Lambda^{-1})^T = W'\mathbf{P}_X(X')\mathbf{P}_X(X)^T. \tag{3}$$

We describe the full derivations for each lemma in the Appendix B.1 and B.2, respectively. $\mathbf{P}_X$ represents a combination of a rotation-reflection matrix ($V$) and a projection-scaling matrix ($\Lambda^{-1}$) obtained from $X$, where $\mathbf{P}_X(X) = XV\Lambda^{-1} = U\Lambda V^T V\Lambda^{-1} = U$ is an orthogonal matrix $U$.

Fig.2 depicts the concept of $\mathbf{P}_X(X)$ and $\mathbf{P}_X(X')$ in the unit hyper-sphere space and each arrow in the hyper-sphere indicates the unique dimensional bases of $X$ and $X'$ such that each contains unique information. If there is modality-specific information that is not shared between $X$ and $X'$, it becomes almost impossible to deduce such information from $X'$ relying solely on the bases of $X$. This consequently lead to an information loss in $\mathbf{P}_X(X')$ within the projected space (refer to the blue arrows in Fig.2-(a)). Hence, Eq.3 suggests that the performance of the student network in cross-modal feature distillation is determined not just by the quality of teacher weights ($W^*$), but also by the amount of transferable general information between the modalities.

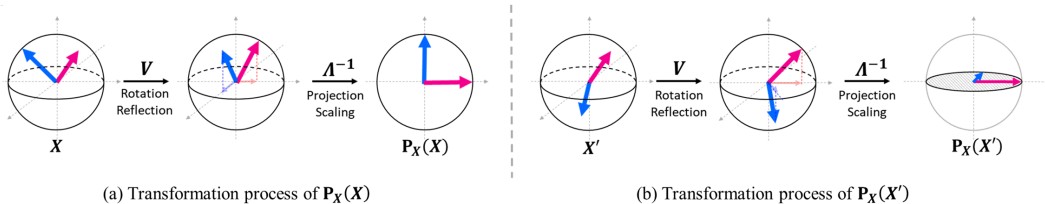

(a) Transformation process of $\mathbf{P}_X(\boldsymbol{X})$          (b) Transformation process of $\mathbf{P}_X(\boldsymbol{X'})$

Figure 2: Concept of $\mathbf{P}_X$ defined in Eq.3. Spheres and circles represent $N$- and $D$-dimensional unit hyper-spheres, respectively. The colored arrows represent the dimensional bases of the $X$ and $X'$.

## 3.4 DIMENSIONAL COLLAPSE CAUSED BY MODALITY GAP

Assume that $X$ and $X'$ share *modality-general parts* along the first $D_g$ dimensions, and each one has *modality-specific parts* along the remaining $D_s$ dimensions (i.e. $D_g + D_s = D$). Then, $X$ and $X'$ can be decomposed as follows:

$$X' = \begin{pmatrix} G \\ S' \end{pmatrix} \quad X = \begin{pmatrix} G \\ S \end{pmatrix}, \tag{4}$$

where $G \in \mathbb{R}^{D_g \times N}$ denotes the modality-general part, and $S, S' \in \mathbb{R}^{D_s \times N}$ denotes the modality-specific parts respectively. In order to thoroughly investigate the impact of modality gap, we assume that $G, S$ and $S'$ will not share any information, which also means that each row of $G, S$ and $S'$ is orthogonal to each other, *i.e.*, $GS^T = O, GS'^T = O$, and $SS'^T = O$ [2]. We will describe this ideal case as $X$ and $X'$ are *completely* separated with $D_g$ shared dimensions.

**Theorem 1.** *In the event that $X$ and $X'$ are completely separated with the shared dimensions $D_g$, the rank of the optimal student weights $W^*$ is bounded by the minimum value between the rank of teacher weight $W'$ and shared dimensionality $D_g$.*

$$\mathrm{rank}(W^*) \leq \min(\mathrm{rank}(W'), D_g). \tag{5}$$

---

[2]In Sec.5, we have empirically demonstrated that our theoretical insights from those assumptions can be extended to complex non-linear and non-separable settings including real-world multi-modal datasets.

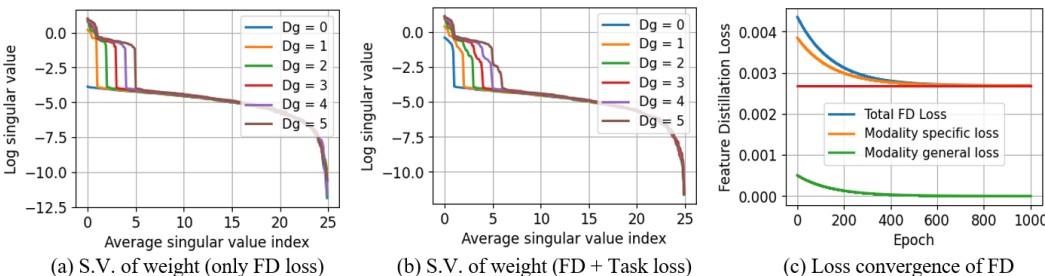

(a) S.V. of weight (only FD loss)    (b) S.V. of weight (FD + Task loss)    (c) Loss convergence of FD

Figure 3: Experimental results on the synthetic dataset. **(a)-(b)** The log singular value spectrum of student's weights learned through distillation, according to the modality general dimension $D_g$. **(c)** The distillation loss dynamics during training, separated into modality-general and modality-specific components. The horizontal red line represents the values derived from Corollary 1.1.

The proof of Theorem 1 can be found in Appendix B.3. Since the rank of the weight directly impacts the dimensionality of the extracted feature, Eq.5 implies that the representation power of the student network during feature distillation is bounded by the quality of the teacher and the shared information across modalities. That is, even if the teacher's feature contains rich information, the limited general information shared between the teacher and student modalities can result in the student features being less representative, which subsequently leads to performance degeneration. We provided more detailed descriptions in the Section 3.3.

**Corollary 1.1.** *If $X$ and $X'$ are completely separated with $D_g$ shared dimensions, Mean Squared Error based feature distillation loss is converged to*

$$L_{FD}(\infty) = \frac{1}{2}\|W'\begin{pmatrix} O \\ S' \end{pmatrix}\|_F^2.$$

(6)

Proof is provided in Appendix B.4. Corollary 1.1 implies that in the completely separated scenario, the teacher-specific information stays entirely distinct from the student data, which hinders the ability to learn any potential patterns that exist in the complex dynamic between two modalities.

### 3.5 VALIDATION ON SYNTHETIC DATASET

Following the experimental setup outlined in (Xue et al., 2022), we assess the validity of our claim on dimensional collapse using a synthetic binary classification dataset, where the size of the modality-general dimensions ($D_g$) can be directly manipulated.

**Singular Value Spectrum (Theorem 1.)**    Here we analyze the distribution of singular values of the student's weights to validate the rank inequality stated in Theorem 1. Fig.3-(a) and (b) show the distribution of log-scaled singular values of the student's weights when only the feature distillation (FD) loss is applied and when both the FD loss and classification task loss are applied simultaneously, respectively. To ensure experimental rigor, we repeated the process ten times and aggregated the singular values for comprehensive analysis. When only FD is applied, as anticipated, we observe a sharp decrease in the singular values after the singular value indices $D_g$. Such abrupt decreases indicate the dimensional collapse of the model. Remarkably, we also observe the same trend in the presence of the task loss. These results suggest that Theorem 1 can be extended to typical cross-modal learning scenarios where both feature distillation and task loss are present.

**Loss Convergence (Corollary 1.1).**    We also examine the training loss dynamics in Fig.3-(c) to verify Corollary 1.1. We observe that the feature distillation (FD) loss converges to specific values calculated from Eq.6, where the modality-general loss asymptotically converges to zero, and the modality-specific loss is bounded by the teacher-specific information $S'$. This implies that our cross-modal distillation strategy should focus on extracting and transferring modality-general information while excluding intractable modality-specific information, as discussed in Sec.3.4.

It also should be noted that the similar results are observed with a cross-entropy distillation loss. We have provided more experimental results and analysis for the synthetic dataset setting in Appendix E.5.

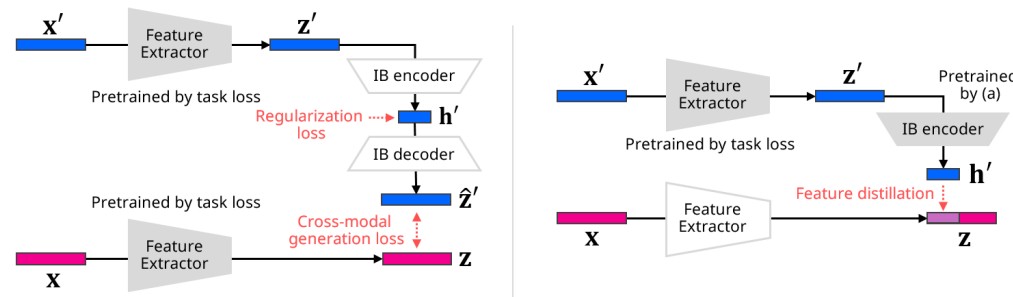

(a) Training information bottleneck model          (b) Sub-dimensional distillation using bottleneck encoder

Figure 4: An overview of CIBA framework. Gray shading represents pretrained and fixed models. (a) To extract modality-general information, both the teacher and student models are trained with task loss. Subsequently, an information bottleneck model is trained to generate target student features from the corresponding teacher's features. (b) Then the learned bottleneck feature $\mathbf{h}'$ is transferred to sub-dimensions of the student feature.

### 3.6 LIMITATIONS AND PRACTICAL EXTENSIONS OF THEORETICAL RESULTS

Our Theorem 1 and Corollary 1.1 suggest that excluding teacher-specific information is crucial when transferring cross-modal knowledge. We are able to transfer only the modality-general information $G$ to the student, as the teacher-specific information $S'$ can be explicitly isolated (Eq.4) and when provided by a linear feature extractor. However, in most real-world applications, $X$ and $X'$ contain a mixture of modality-general, modality-specific, and sensor noises (Hälvä et al., 2021). In other words, $X$ and $X'$ are *not completely separable*. Moreover, modern feature extractors employ nonlinear layers (Hyvärinen et al., 2023), hence it is extremely difficult to completely extract only the modality-general information from the teacher. To extend our theoretical findings to such nonlinear and practical applications, we propose a method to approximate modality-general information for CMFD (Sec.4), and validate its effectiveness on various real-world multi-modal datasets (Sec.5).

## 4 METHODOLOGY

### 4.1 EXTRACTING MODALITY-GENERAL FEATURES

Real-world data includes both modality-general and specific information as well as noise (*e.g.*, sensor noise). Moreover, nonlinearity of modern neural network models makes it even more difficult to disentangle such compounded information from the learned features (Chartsias et al., 2020; Liu et al., 2022). The *Information Bottleneck* principal (Alemi et al., 2016; Tishby et al., 1999) can be a promising solution, as it tries to learn concise, disentangled representations from the input data while eliminating irrelevant information to the target data. Motivated by this concept, we introduce the **C**ross-modal **I**nformation **B**ottleneck **A**pproximation (**CIBA**) framework to extract modality-general features for effective CMFD. Fig.4 depicts an overview of CIBA. In particular, CIBA aims to extract a sub-dimensional representation of modality-general features by minimizing modality-specific information through an encoder-decoder structure as illustrated in Fig.4-(a).

Suppose that $D$-dimensional features, denoted as $\mathbf{z}'$ and $\mathbf{z}$ are obtained from teacher and student models trained without knowledge distillation, respectively. Defining the encoder as $p_\theta$, the decoder as $q_\phi$, and the $H$-dimensional bottleneck feature as $\mathbf{h}'$ ($H \leq D$), the optimization objective for extracting modality-general feature is calculated by:

$$L_{IB} = -\mathbb{E}_q[\log q_\phi(\mathbf{z}|\mathbf{h}')] + \mathcal{D}_{KL}(p_\theta(\mathbf{h}'|\mathbf{z}'), p(\mathbf{h}')), \tag{7}$$

where $\mathcal{D}_{KL}$ denotes Kullback-Leibler divergence. Here the first term is a cross-modal generation loss, which takes the teacher model's feature $\mathbf{z}'$ as input and aims to make the decoded output $\hat{\mathbf{z}}'$ similar to the student model's feature $\mathbf{z}$. This term encourages the bottleneck feature $\mathbf{h}'$ to preserve the modality general information. The second term regularizes $\mathbf{h}'$, encouraging the elimination of teacher-specific information. We formulated the cross-modal generation loss as the L2-distance between $\hat{\mathbf{z}}'$ and $\mathbf{z}$, and assumed an isotropic Gaussian distribution for $p(\mathbf{h})$ and the encoder's output, allowing closed-form calculation (Alemi et al., 2016; Kingma & Welling, 2013). Finally, we take the bottleneck feature $\mathbf{h}'$ for cross-modal knowledge distillation.

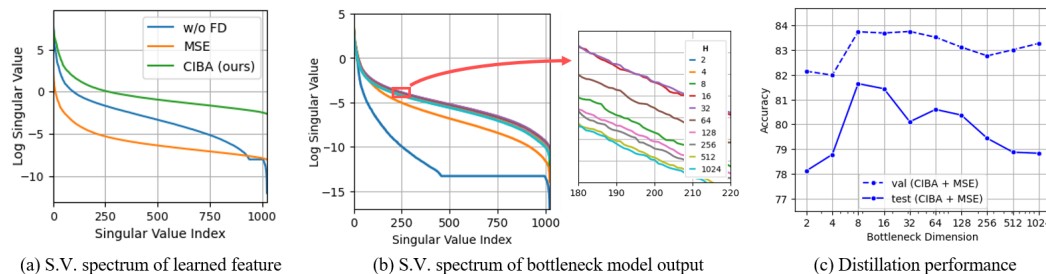

(a) S.V. spectrum of learned feature     (b) S.V. spectrum of bottleneck model output     (c) Distillation performance

Figure 5: Experimental results on RAVDESS. **(a)** The log singular value spectrum of the learned student's feature with and without distillation. **(b)** The log singular value spectrum of the bottleneck model's output depending on the dimension of bottleneck feature $H$. **(c)** The performance trend of CIBA implemented with the sub-dimensional distillation using MSE, depending on $H$.

### 4.2 SUB-DIMENSIONAL FEATURE DISTILLATION

Although we have the concise representation of modality-general features $\mathbf{h}' \in \mathbb{R}^H$, there still exists a dimensionality mismatch issue between $\mathbf{h}'$ and student features $\mathbf{z} \in \mathbb{R}^D$ (*i.e.*, $H \neq D$). A recent study (Jing et al., 2021) exploited sub-dimensional representations to prevent the dimensional collapse in self-supervised contrastive learning. Inspired by this, we transfer the bottleneck feature $\mathbf{h}'$ into $H$ sub-dimensions of the student features $\mathbf{z}$, as illustrated in Fig.4-(b). Such a sub-dimensional distillation strategy allows the remaining feature dimensions to exclusively learn modality-specific knowledge as shown in Fig.1-(d). Pseudo codes for CIBA framework are provided in the Appendix C.

## 5 EXPERIMENT

We validate the effectiveness of CIBA scheme on four real-world datasets, including RAVDESS (Livingstone & Russo, 2018), MM-IMDB (Arevalo et al., 2017), nuScenes (Caesar et al., 2020), and VGG-Sound (Chen et al., 2020). Implementation details are provided in the Appendix D.

### 5.1 RAVDESS (AUDIO-IMAGE)

#### 5.1.1 DIMENSIONAL COLLAPSE IN FEATURE SPACE

We first demonstrate that our Claim 1, which pertains to the distillation of teacher-specific information inducing dimensional collapse, also holds valid for real-world datasets and non-linear feature extractors. In Fig.5-(a), we compared the singular value spectrum of features from the student model (image baseline) trained without knowledge distillation ("w/o FD"), trained with a baseline strategy ("MSE"), and our bottleneck scheme ("CIBA"). The features trained with MSE exhibit lower singular values compared to the image baseline, indicating dimensional collapse of the feature space. Conversely, the features trained with CIBA exhibit larger singular values compared to the student baseline.

Similar trend echoed in Fig.1, where features trained with MSE (Fig.1-(c)) only span modality-general information, leading to a lack of dimensional diversity. In contrast, features learned through our distillation strategy (Fig.1-(d)) can represent not only modality-general information, but also student-specific information. These results extend our claim to real-world scenarios.

#### 5.1.2 KNOWLEDGE DISTILLATION EFFICACY

To evaluate the effectiveness of CIBA framework, in Tab.1-(a), we compared it with the various distillation strategy including MSE, cross-entropy (CE) (Kwon et al., 2020), CLIP (Radford et al., 2021), and margin loss (Jin et al., 2023). MSE leads the student features to be biased towards modality-general information, as illustrated in Fig.1-(c), resulting in a marginal improvement of 0.34% in test performance compared to the image baseline. In contrast, CIBA leads to a substantial improvement of 3.56% in test performance (MSE+CIBA) by effectively leveraging both modality-general information and image-specific details, as illustrated in Fig.1-(d). Similar trends are observed with other distillation losses. These results demonstrate the effectiveness of CIBA in extracting and sub-dimensional transferring modality-general information.

Table 1: Results on RAVDESS and MM-IMDB. Modality 'A', 'I', and 'T' denote Audio, Image, and Text respectively. The number within the parentheses denotes the dimension of the bottleneck feature. Further statistical analyses, including p-value analysis and box plots for the results, are provided in the Appendix E.1.

(a) RAVDESS dataset

| Method | Modality | Val. | Test |
|---|---|---|---|
| Audio-baseline | A | $73.26_{1.51}$ | $72.22_{2.26}$ |
| Image-baseline | I | $81.64_{0.79}$ | $78.08_{1.30}$ |
| MSE | $A \rightarrow I$ | $80.91_{0.68}$ | $78.42_{0.69}$ |
| MSE + CIBA (8) | $A \rightarrow I$ | $83.74_{0.98}$ | $81.64_{1.38}$ |
| CE | $A \rightarrow I$ | $82.17_{0.70}$ | $78.83_{0.93}$ |
| CE + CIBA (8) | $A \rightarrow I$ | $83.22_{0.51}$ | $81.19_{1.58}$ |
| CLIP | $A \rightarrow I$ | $81.33_{1.23}$ | $78.79_{1.53}$ |
| CLIP + CIBA (64) | $A \rightarrow I$ | $83.80_{0.39}$ | $80.77_{1.04}$ |
| Margin | $A \rightarrow I$ | $81.82_{0.81}$ | $78.62_{1.24}$ |
| Margin + CIBA (128) | $A \rightarrow I$ | $83.46_{0.94}$ | $80.42_{1.14}$ |

(b) MM-IMDB datatset

| Method | Modality | F1-micro | F1-macro |
|---|---|---|---|
| Image-baseline | I | $40.00_{0.50}$ | $25.82_{0.63}$ |
| Text-baseline | T | $57.87_{0.27}$ | $45.95_{0.38}$ |
| Fusion-baseline | F (I+T) | $57.09_{0.41}$ | $46.10_{0.41}$ |
| MSE | $I \rightarrow T$ | $56.74_{0.22}$ | $42.18_{0.48}$ |
| +CIBA w/ DVIB (16) | $I \rightarrow T$ | $59.29_{0.16}$ | $47.71_{0.22}$ |
| +CIBA w/ SA (16) | $I \rightarrow T$ | $58.86_{0.54}$ | $46.51_{0.66}$ |
| +CIBA w/ VQ-VAE (16) | $I \rightarrow T$ | $58.67_{0.45}$ | $46.71_{0.37}$ |
| MSE | $F \rightarrow T$ | $58.32_{0.28}$ | $45.77_{0.16}$ |
| +CIBA w/ DVIB (32) | $F \rightarrow T$ | $58.73_{0.38}$ | $47.11_{0.46}$ |
| +CIBA w/ SA (32) | $F \rightarrow T$ | $58.46_{0.16}$ | $46.19_{0.34}$ |
| +CIBA w/ VQ-VAE (32) | $F \rightarrow T$ | $58.48_{0.18}$ | $46.44_{0.33}$ |

### 5.1.3 OPTIMAL BOTTLENECK DIMENSION

The dimension ($H$) of bottleneck feature $\mathbf{h}'$ in Eq.7 defines the conciseness of the modality-general features generated from the bottleneck model as shown in Fig.4. When $H$ is too small, it may not effectively compress all modality-general information, resulting in insufficient capture of the target student features. Conversely, with too large $H$, the bottleneck features may contain not only modality-general information but also irrelevant teacher-specific information and noise, which could degrade distillation performance. Hence, it is crucial to find the optimal value of $H$ that satisfies both criteria.

To assess the impact of $H$ on distillation performance, we conducted experiments by gradually increasing $H$ with $k$ ranging from 1 to 10, using increments of $2^k$, given that both teacher and student features have a dimensionality of 1024 in the RAVDESS setting. First, we present the singular value spectrum of the learned features from the bottleneck model (*i.e.*, $\hat{\mathbf{z}}'$), varying with $H$, to evaluate the quality of extracted bottleneck features $\mathbf{h}'$ in Fig.5-(b). The spectrum exhibits a nearly identical distribution for $H \geq 8$, suggesting that 8-dimensional bottleneck features possess adequate capacity to capture the general information required for enhancing quality of the outputs from the bottleneck decoder. Fig.5-(c) illustrates the relationship between $H$ and distillation performance, with superior performance observed at $H = 8$. These findings align with the analyses of the singular value spectrum. Thus, we may conclude that the conciseness ($H$) of modality-general bottleneck features $\mathbf{h}'$ crucially impacts distillation performance. It is noteworthy that additional distillation constraints from methods such as CLIP and Margin loss can disrupt the full transmission of bottleneck feature information. Imposing a larger $H$ can reduce this disruption, as shown in Tab.1-(a).

### 5.2 MM-IMDB (IMAGE-TEXT)

### 5.2.1 FUSION MODEL AS TEACHER

According to the results in Tab.1-(b), due to the limited representational power of the image features, MSE-based distillation from the image teacher leads to a degradation in performance. Additionally, distillation from the fusion teacher also lead to only a marginal improvement compared to the text baseline. While the fusion model can extract more modality-general features by utilizing both modalities (Xue et al., 2022), it may still contain noisy and image-specific knowledge.

Therefore, as shown in Tab.1-(b), CIBA strategy can improve the distillation performance for both the fusion teacher and image teacher by effectively removing such image-specific and noisy information. In the Fig.6, CIBA exhibits the larger singular values spectrum of learned students for both fusion teacher and image teacher, indicating an increase in feature representation power. A noteworthy observation is that the optimal bottleneck dimension for the fusion teacher ($H = 32$) is greater than that of the image teacher ($H = 16$). This finding is reasonable since features of fusion teacher encompass more modality general information, necessitating a larger bottleneck dimensions for effective representation. In Appendix E.3, we also present fusion teacher experiments conducted on the RAVDESS dataset, demonstrating similar performance improvements.

### 5.2.2 ABLATION OF BOTTLENECK STRUCTURE

We adopt the DVIB structure from (Alemi et al., 2016) as a representative bottleneck model. However, any types of encoder-decoder-based bottleneck structures can be applied to our CIBA framework. In this ablation study, we perform the distillation process with two models: 1) **Self-Attention (SA)** and 2) **VQ-VAE**. Self-Attention is based on (Srinivas et al., 2021), in which multi-head self-attention module in Transformer (Vaswani et al., 2017) is applied between encoder and decoder architecture. Also, inspired by the concept of DVIB that modifies the self-reconstruction term of VAE (Kingma & Welling, 2013), VQ-VAE employs the structure of the vector quantized VAE (Van Den Oord et al., 2017), adjusting the loss term from self-reconstruction loss to cross-generation loss. The results presented in Tab.1-(b) demonstrate that proposed models outperform all MSE and text baseline models. This demonstrates that the information bottleneck scheme effectively extracts modality-general information.

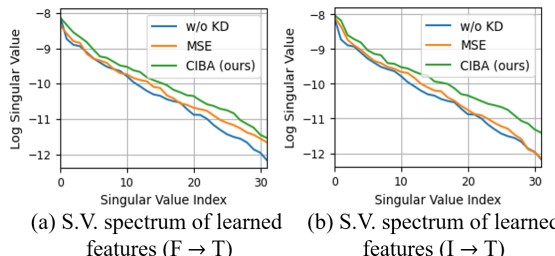

(a) S.V. spectrum of learned features (F → T)  (b) S.V. spectrum of learned features (I → T)

Figure 6: Evaluation of the proposed scheme implemented by DVIB, for both the fusion-to-text and the image-to-text scenarios on the MM-IMDB. **(a)** and **(b)** present the singular value spectrum of the learned student's feature with and without distillation.

### 5.3 NUSCENES (LIDAR-CAMERA)

We extend the validation of our method to the challenging 3D object detection (3DOD) task with the nuScenes benchmark (Caesar et al., 2020). Following prior works on CMKD for 3DOD (Chen et al., 2023; Hong et al., 2022; Li et al., 2022b), we adopt a LiDAR-based model (Wang & Solomon, 2021) as the teacher and a camera-based multi-view model (Li et al., 2022e) as the student. These prior works broadly utilize the MSE-like feature distillation strategy, while various output-level knowledge distillation approaches are also introduced to enhance the distillation performance. To examine the sole effect of our feature distillation strategy (CIBA), we exclusively utilized MSE (Hong et al., 2022; Li et al., 2022c) and MSE around the ground truth (MSE w/ GT) (Chen et al., 2023) losses as the baseline for comparison.

Tab.2 depicts that only applying MSE-like losses results in a minor improvement of up to 0.6 of NDS, while the introduction of CIBA strategy further im-

Table 2: Results on nuScenes. Modality 'L' and 'I' denote LiDAR and Image, respectively. The number within the parentheses denotes the dimension of the bottleneck feature.

| Method | Modality | NDS(%) |
|---|---|---|
| Obj-DGCNN | L | 66.7 |
| BEVFormer | I | 43.4 |
| MSE | L → I | 43.8 |
| MSE w/ GT | L → I | 44.0 |
| MSE + CIBA (2) | L → I | 44.4 |
| MSE + CIBA (4) | L → I | **44.8** |
| MSE + CIBA (8) | L → I | 44.4 |
| MSE + CIBA (16) | L → I | 44.0 |
| MSE + CIBA (32) | L → I | 43.7 |
| MSE + CIBA (64) | L → I | 42.1 |
| MSE + CIBA (128) | L → I | 39.8 |

proves the distillation performance by upto $1.4\%$ compared to the camera-based student baseline. Given the challenging nature of camera-based 3DOD, which involves predicting object classes and bounding box parameters (center position, box dimensions, and orientation) without precise depth cues, such improvements can be considered to be significant.

Additionally, the best performance is achieved when $H = 4$, and we are surprised at such a small number of effective dimension compared to the original feature dimension of $256$. Camera images contain semantically rich information such as colors and textures, while LiDAR point clouds provide precise depth cues. Hence, we speculate that the potentially sharable modality-general features could be the shape information and those information can be sufficiently represented by only small portion of the feature space. These results extend our claim on dimensional collapse to the challenging 3DOD tasks, indicating that transferring sub-dimensional modality-general features could be more beneficial than MSE-like global feature distillation.

Table 3: Results on VGG-Sound. Modality 'V' and 'A' denote Video and Audio, respectively. Subscript '50' and '18' under the modality symbol indicate that the ResNet-50 and ResNet-18 models are employed as backbone models. The number within the parentheses denotes the dimension of the bottleneck feature. Further statistical analyses, including p-value analysis are provided in Appendix E.2.

| Method | Modality | Val. | Test |
|---|---|---|---|
| Video-baseline (ResNet-50) | $V_{50}$ | $50.42_{\ 0.37}$ | $49.43_{\ 0.66}$ |
| Audio-baseline (ResNet-50) | $A_{50}$ | $69.55_{\ 0.36}$ | $68.76_{\ 0.33}$ |
| Video-baseline (ResNet-18) | $V_{18}$ | $42.11_{\ 0.53}$ | $41.53_{\ 0.41}$ |
| Audio-baseline (ResNet-18) | $A_{18}$ | $68.86_{\ 0.33}$ | $69.08_{\ 0.52}$ |
| MSE | $V_{18}{\rightarrow}A_{18}$ | $67.54_{\ 0.76}$ | $68.32_{\ 0.49}$ |
| MSE + CIBA (16) | $V_{18}{\rightarrow}A_{18}$ | $70.11_{\ 0.40}$ | $70.39_{\ 0.48}$ |
| MSE | $V_{50}{\rightarrow}A_{18}$ | $68.53_{\ 0.36}$ | $68.54_{\ 0.34}$ |
| MSE + CIBA (16) | $V_{50}{\rightarrow}A_{18}$ | $70.21_{\ 0.28}$ | $70.71_{\ 0.52}$ |
| MSE | $A_{18}{\rightarrow}V_{18}$ | $42.61_{\ 0.33}$ | $41.28_{\ 0.57}$ |
| MSE + CIBA (16) | $A_{18}{\rightarrow}V_{18}$ | $43.59_{\ 0.57}$ | $42.55_{\ 0.62}$ |
| MSE | $A_{50}{\rightarrow}V_{18}$ | $41.40_{\ 0.71}$ | $40.33_{\ 0.39}$ |
| MSE + CIBA (16) | $A_{50}{\rightarrow}V_{18}$ | $43.44_{\ 0.33}$ | $42.95_{\ 0.19}$ |

## 5.4 VGG-Sound (Video-Audio)

We also validated the proposed method on the video event classification dataset, VGG-Sound (Chen et al., 2020). Referring to prior work (Xue et al., 2022), we employed ResNet models (He et al., 2016) for both modalities and utilized the features extracted after the average pooling layer for the distillation. To further validate our approach with varying levels of encoder capacity, we adopted both ResNet-18 and ResNet-50 models as teacher models.

Tab.3 presents the experimental results, which are consistent with those observed on other datasets (Tab.1 and 2). The MSE-based approach, which propagates both modality-general and modality-specific information, results in only marginal performance improvements over the baseline model or even degrades performance. In contrast, the proposed method consistently achieved significant improvements in distillation performance across all scenarios. These findings demonstrate that our investigation and the proposed approach remain effective even in the large-scale dataset.

In addition, to investigate the impact of the bottleneck dimension $H$ on a large-scale dataset, we conducted extensive ablation experiments. Detailed results are provided in Appendix E.7, and the results align with the analysis presented in Sec.5.1.3 and Fig.5. Specifically, for most $H$ values except for a few extreme cases, the proposed method consistently outperformed the MSE approach, regardless of the backbone's capacity. Furthermore, applying the method described in Sec.5.1.3 to select an adequate $H$ yielded $H = 16$, as shown in the upper plot of Fig.12 in Appendix E.7. The results for $H = 16$ consistently demonstrated sufficiently strong performance, as illustrated in the lower plot of Fig.12 in Appendix E.7. These findings highlight the significant potential of the proposed method for practical applications in real-world scenarios.

## 6 Discussion and Conclusion

In this paper, we investigate the impact of distributional shifts between teacher and student modalities in cross-modal feature distillation. We first theoretically validate that transferring modality-specific information from the teacher model, which is intractable for the student, leads to dimensional collapse in the learned student features, resulting in degraded distillation quality. Then, we also demonstrate our claim on dimensional collapse using a synthetic dataset. To minimize the adversarial impact of the modality gap, we propose the cross-modal information bottleneck approximation (CIBA) framework for cross-modal feature distillation. Our approach aims to extract modality-general features from the teacher and distill them to sub-dimensions of student features. We validate the effectiveness of CIBA on various real-world multi-modal datasets, including audio-visual (RAVDESS), image-text (MM-IMDB), and image-point clouds (nuScenes). In addition to feature-level distillation, As future work, we plan to further explore output-level distillation in cross-modal knowledge distillation and to study their interplay through both theoretical and empirical analyses. Additionally, we aim to extend our theoretical contributions to more challenging assumptions, such as a non-linear feature extractor.

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

# A USEFUL LEMMAS

**Lemma 3.** *Negative gradient of student weight $W$ is calculated by linear transform of $W$:*

$$-\frac{\partial L_{FD}}{\partial W} = AW^T + B$$

$$A = -XX^T, \quad B = XX'^T W'^T$$

$$(8)$$

*Proof.* Let us define $\mathbf{z}_i = W\mathbf{x}_i$ and $\mathbf{z}_i' = W'\mathbf{x}_i'$, respectively. Then gradient of $W$ can be obtained using the chain rule.

$$\frac{\partial L_{FD}}{\partial W} = \sum_{i=1}^N \frac{\partial \mathbf{z}_i}{\partial W} \cdot \frac{1}{2} \cdot \frac{\partial \|\mathbf{z}_i' - \mathbf{z}_i\|_2^2}{\partial \mathbf{z}_i}. \tag{9}$$

Eq.9 can be easily calculated, as the first term on the right-hand side is the derivative of the linear equation $\mathbf{z}_i = W\mathbf{x}_i$, and the second term on the right-hand side is the derivative of the squared L2-norm.

$$\frac{\partial \mathbf{z}_i}{\partial W} = \frac{\partial W\mathbf{x}_i}{\partial W} = \mathbf{x}_i \tag{10}$$

$$\frac{1}{2} \cdot \frac{\partial \|\mathbf{z}_i' - \mathbf{z}_i\|_2^2}{\partial \mathbf{z}_i} = (\mathbf{z}_i - \mathbf{z}_i')^T. \tag{11}$$

Thus the gradient of $W$ can be obtained by

$$\frac{\partial L_{FD}}{\partial W} = \sum_{i=1}^N \mathbf{x}_i(\mathbf{z}_i - \mathbf{z}_i')^T. \tag{12}$$

With gradient descent optimization, weight is updated by

$$
\begin{aligned}
-\frac{\partial L_{FD}}{\partial W} &= \sum_{i=1}^N -\mathbf{x}_i(\mathbf{z}_i - \mathbf{z}_i')^T \\
&= \sum_{i=1}^N -\mathbf{x}_i(W\mathbf{x}_i - W'\mathbf{x}_i')^T \\
&= \sum_{i=1}^N \left( -\mathbf{x}_i\mathbf{x}_i^T W^T + \mathbf{x}_i\mathbf{x}_i'^T W'^T \right) \\
&= -XX^T W^T + XX'^T W'^T \\
&= AW^T + B.
\end{aligned}
\tag{13}
$$

$\square$

# B PROOFS

## B.1 PROOF OF LEMMA 1

We provide two proofs for Lemma 1. The first approach achieves the optimal solution by leveraging the convex property of the MSE loss. The second approach is more general, obtaining the converged solution through gradient updates. When task loss or other loss functions are added, the convexity of the loss function may not hold. Therefore, in more complex situations, the second approach can be useful for analyzing the converged weight values.

*Proof 1 (Use convex property).* Since MSE loss in Eq.1 is a convex function with respect to $W$, the global optimal solution, denoted as $W^*$, corresponds to the point where the gradient of $W$ becomes zero. According to Eq.12 and Eq.13 in Lemma 3, the gradient of $W$ is represented by

$$\frac{\partial L_{FD}}{\partial W} = XX^T W^T - XX'^T W'^T. \tag{14}$$

Then optimal solution $W^*$ should satisfy the zero-gradient condition.

$$\frac{\partial L_{FD}}{\partial W}\bigg|_{W=W^*} = XX^T W^{*T} - XX'^T W'^T = [0]_{F \times D}. \tag{15}$$

Since we assumed that $X$ is a fully-ranked matrix, $XX^T \in \mathbb{R}^{D \times D}$ is also fully-ranked matrix. Thus $XX^T$ is invertible. Then Eq.15 can be developed as

$$W^{*T} = (XX^T)^{-1} XX'^T W'^T$$
$$W^* = W'X'X^T(XX^T)^{-1}. \tag{16}$$

$\square$

*Proof 2 (Gradient update).* Eq.8 in Lemma 3 is non-homogeneous linear differential equation. For notation simplicity, omit transpose mark of $W$ in Eq.8 and add iteration parameter $t \in [0, \infty]$. Then solution for differential equation can be calculated as

$$W(t) = AW(t) + B$$
$$= AW(t) + AA^{-1}B$$
$$= A(W(t) + A^{-1}B)$$
$$= A(W(t) + W^*) \quad \leftarrow \quad W^* := A^{-1}B \tag{17}$$
$$\text{(Trivial solution of homogeneous differential equation)}$$
$$\dot{W}(t) = e^{At}(W(0) - W^*) + W^*, \quad where \quad W^* = -A^{-1}B.$$

Then trivial solution of Eq.17 can be derived by

$$W(t) = e^{At}(W(0) - W^*) + W^*, \quad where \quad W^* = -A^{-1}B. \tag{18}$$

Since $A = -XX^T$ is negative definite, $e^{At}$ goes to zero as $t \to \infty$. Therefore student weight $W(t)$ is converged to $W^*$ with gradient descent optimization.

$$W(\infty)^T = W^{*T} = -A^{-1}B$$
$$W(\infty) = W^* = -B^T A^{-1T}. \tag{19}$$

$\square$

### B.2 PROOF OF LEMMA 2

*Proof.* Since $XX^T$ is real-symmetric and fully-ranked matrix (*i.e.* symmetric positive definite), $XX^T$ can be decomposed by

$$XX^T = (U\Lambda V^T)(V\Lambda^T U^T)$$
$$X = U\Lambda V^T \text{(by singular value decomposition)}$$
$$U^T = U^{-1} \text{(Unitary matrix)}$$
$$V^T = V^{-1} \text{(Unitary matrix)} \tag{20}$$
$$\Lambda = \begin{pmatrix} \sigma_1 & \dots & 0 & \dots & 0 \\ \vdots & \ddots & \vdots & \dots & 0 \\ 0 & \dots & \sigma_D & \dots & 0 \end{pmatrix}.$$

where $(\sigma_1, ..., \sigma_D)$ are positive singular values of $X$.

Before developing Eq.16, define $\Lambda^{-1}$, right inverse matrix of $\Lambda$ satisfying $\Lambda\Lambda^{-1} = \mathbf{I}_{D \times D}$.

$$\Lambda^{-1} = \begin{pmatrix} 1/\sigma_1 & \dots & 0 \\ \vdots & \ddots & \vdots \\ 0 & \dots & 1/\sigma_D \\ \vdots & \vdots & \vdots \\ 0 & \dots & 0 \end{pmatrix}. \tag{21}$$

Then Eq.20 can be expanded by

$$
\begin{aligned}
W^* &= W'X'X^T(XX^T)^{-1} \\
&= W'X'(V\Lambda^T U^T)(U\Lambda\Lambda^T U^T)^{-1} \\
&= W'X'(V\Lambda^T U^T U)(\Lambda\Lambda^T)^{-1}U^T \\
&= W'X'V\Lambda^T(U^T U)(\Lambda^{-T}\Lambda^{-1})U^T \\
&= W'X'V(\Lambda^T\Lambda^{-T}\Lambda^{-1})U^T \\
&= W'X'V\Lambda^{-1}U^T \\
&= W'(X'V\Lambda^{-1})\mathbf{I}_{D\times D}U^T \\
&= W'(X'V\Lambda^{-1})(\Lambda^{-T}\Lambda^T)U^T \\
&= W'(X'V\Lambda^{-1})(\Lambda^{-T}V^T V\Lambda^T)U^T \\
&= W'(X'V\Lambda^{-1})(\Lambda^{-T}V^T)(V\Lambda^T U^T) \\
&= W'(X'V\Lambda^{-1})(\Lambda^{-T}V^T)X^T \\
&= W'(X'V\Lambda^{-1})(XV\Lambda^{-1})^T.
\end{aligned}
\tag{22}
$$

Let us define a transformation $\mathbf{P}_X(M) = MV\Lambda^{-1}$. Then we can develop Eq.22 as below.

$$
\begin{aligned}
W^* &= W'(X'V\Lambda^{-1})(XV\Lambda^{-1})^T \\
&= W'\mathbf{P}_X(X')\mathbf{P}_X(X)^T.
\end{aligned}
\tag{23}
$$

$\square$

### B.3   Proof of Theorem 1

From Lemma 2, $W$ can be written as below.

$$
\begin{aligned}
W^* &= W'(X'V\Lambda^{-1})(\Lambda^{-T}V^T X) \\
&= W'\mathbf{P}_X(X')\mathbf{P}_X(X)^T.
\end{aligned}
\tag{24}
$$

Since $\mathbf{P}_X$ is combination of rotation, reflection, scaling, and linear projection matrices, $\mathbf{P}_X$ can be applied to sub-matrix $G$, $S$ and $S'$ individually.

$$
\begin{aligned}
W^* &= W'\mathbf{P}_X(X')\mathbf{P}_X(X)^T \\
&= W'\begin{pmatrix} \mathbf{P}_X(G) \\ \mathbf{P}_X(S') \end{pmatrix}\begin{pmatrix} \mathbf{P}_X(G)^T & \mathbf{P}_X(S)^T \end{pmatrix}.
\end{aligned}
\tag{25}
$$

By the complete separation assumption in Eq.4, each row of $G$, $S$ and $S'$ are orthogonal, and $\mathbf{P}_X$ is combination of rotation, reflection, scaling, and linear projection matrices. Thus $\mathbf{P}_X(G)$, $\mathbf{P}_X(S)$ and $\mathbf{P}_X(S')$ are still orthogonal each other. In addition, since $\mathbf{P}_X(X)$ is unitary matrix, $\mathbf{P}_X(G)\mathbf{P}_X(G)^T = \mathbf{I}_{D_g\times D_g}$. Then Eq.25 can be developed as

$$
\begin{aligned}
W^* &= W'\begin{pmatrix} \mathbf{P}_X(G) \\ \mathbf{P}_X(S') \end{pmatrix}\begin{pmatrix} \mathbf{P}_X(G)^T & \mathbf{P}_X(S)^T \end{pmatrix} \\
&= W'\begin{pmatrix} \mathbf{I}_{D_g\times D_g} & O \\ O & O \end{pmatrix}.
\end{aligned}
\tag{26}
$$

Therefore, rank bound of $W^*$ can be obtained as below.

$$
\begin{aligned}
\mathrm{rank}(W^*) &= \mathrm{rank}(W'\begin{pmatrix} \mathbf{I}_{D_g\times D_g} & O \\ O & O \end{pmatrix}) \\
&\leq \min(\mathrm{rank}(W'), \mathrm{rank}(\begin{pmatrix} \mathbf{I}_{D_g\times D_g} & O \\ O & O \end{pmatrix})) \\
&= \min(\mathrm{rank}(W'), D_g).
\end{aligned}
\tag{27}
$$

### B.4    PROOF OF COROLLARY 1.1

*Proof.*  Utilizing the Eq. 27, let us compute the difference between features of the teacher and the learned student.

$$W'X' - W^*X = W'\begin{pmatrix} G \\ S' \end{pmatrix} - W'\begin{pmatrix} \mathbf{I}_{D_g \times D_g} & O \\ O & O \end{pmatrix}\begin{pmatrix} G \\ S \end{pmatrix}$$

$$= W'\begin{pmatrix} G \\ S' \end{pmatrix} - W'\begin{pmatrix} G \\ O \end{pmatrix} \tag{28}$$

$$= W'\begin{pmatrix} O \\ S' \end{pmatrix}.$$

Therefore, $L_{FD}$ is bounded by

$$L_{FD} = \frac{1}{2} \cdot \|W'X' - W^*X\|_F^2$$

$$= \frac{1}{2} \cdot \|W'\begin{pmatrix} O \\ S' \end{pmatrix}\|_F^2. \tag{29}$$

$\square$

## C PSEUDO ALGORITHM OF THE PROPOSED DISTILLATION METHOD

Algorithm 1 describes the training process of the encoder-decoder structured bottleneck model, which is designed to extract modality-general features from the teacher model, as presented in Sec.4.1. We adopt the L2-distance for the cross-modal generation loss, and assume an isotropic Gaussian distribution for both $p(\mathbf{h})$ and the encoder's output, allowing closed-form calculation (Alemi et al., 2016; Kingma & Welling, 2013).

Algorithm 2 describes the sub-dimensional feature distillation method presented in Sec.4.2, utilizing MSE loss as the sub-dimensional distillation loss. Notation $[0:H]$ in Algorithm 2 denotes the first $H$ dimensions of the vector.

---

**Algorithm 1** Pseudo Code for Training Information Bottleneck Model

---

**Input:**
$f'$: Feature extractor of the teacher model pretrained by task loss.
$(f_0, g_0)$: Initial state of feature extractor and task head of the student model.
$(e_0, d_0)$: Initial state of encoder and decoder of the deep variational information bottleneck.
$(X', X, Y)$: Dataset of teacher modality, student modality, and their label set.
$\lambda$: Balancing parameter for the regularization loss.
**Functions:**
UNIMODAL$(f, g, X, Y)$: Train both feature extractor $f$ and task head $g$ with training data $X$ and their label set $Y$
OPTIMIZE$(e, d, l)$: Update the parameters of bottleneck encoder $e$ and decoder $d$ by the gradient descent, given the loss $l$
SUM$(\mathbf{z})$: Calculate the sum of all elements in the vector $\mathbf{z}$
**Procedure:**
1: $f_u, g_u = $ UNIMODAL$(f_0, g_0, X, Y)$
2: $Z', Z = f'(X'), f_u(X)$
3: $(e, d) = (e_0, d_0)$
4: **while** $(e, d)$ is not converged **do**
5:     **for** $(\mathbf{z}'_i, \mathbf{z}_i)$ in $(Z', Z)$ **do**
6:        $\mathbf{h}'_\mu, \mathbf{h}'_\sigma = e(\mathbf{z}'_i)$
7:        $\hat{\mathbf{z}}' = d(\mathbf{h}'_\mu, \mathbf{h}'_\sigma)$
8:        $l_{gen} = \|\hat{\mathbf{z}}' - \mathbf{z}_i\|_2^2$      ▷ Cross-modal generation loss of Eq.7
9:        $l_{reg} = -0.5 \cdot $ SUM$(1 + \log \mathbf{h}'_\sigma - \mathbf{h}'^2_\mu - \mathbf{h}'_\sigma)$     ▷ Regularization loss of Eq.7
10:       $l_i = l_{gen} + \lambda \cdot l_{reg}$
11:     **end for**
12:    $(e, d) = $ OPTIMIZE$(e, d, \sum_i l_i)$
13: **end while**
14: **Return** $(e, d)$

---

**Algorithm 2** Pseudo Code for Calculating Sub-dimensional Feature Distillation Loss

---

**Input:**
$f'$: Feature extractor of the teacher model pretrained by task loss.
$e$: Bottleneck encoder trained using Algorithm 1
$(f, g)$: Feature extractor and task head of the student model.
$(X', X, Y)$: Dataset of teacher modality, student modality, and their label set.
$H$: Dimension of bottleneck feature.
**Procedure:**
1: $Z', Z = e(f'(X')), f(X)$     ▷ Obtaining bottleneck feature from teacher model.
2: **for** $(\mathbf{h}'_i, \mathbf{z}_i)$ in $(Z', Z)$ **do**
3:     $l_i = \|\mathbf{h}'_i - \mathbf{z}_{i[0:H]}\|_2^2$      ▷ Sub-dimensional distillation loss for the fist $H$ dimensions of $\mathbf{z}_i$.
4: **end for**
5: **Return** $\sum_i l_i$

---

# D    IMPLEMENTATION DETAILS

## D.1    SYNTHETIC DATASET

We followed the experimental settings outlined in (Xue et al., 2021), with some modifications. To create completely separable data distributions, we applied the Gram-Schmidt process to the data generated from the original unit normal Gaussian distribution. We also utilized a two-layer linear model without any non-linear activation. The first linear layer acts as the feature extractor, while the second linear layer functions as the classifier. If only the feature distillation loss (Eq.1) is applied without the task loss, the second layer is ignored, thus satisfying our assumed single linear layer setting in Sec.3.2. All models were trained and evaluated on Intel(R) Xeon(R) CPU E5-2620 v3.

## D.2    RAVDESS

The Ryerson Audio-Visual Database of Emotional Speech and Song (RAVDESS) (Livingstone & Russo, 2018) comprises videos featuring professional actors vocalizing sentences with eight distinct emotions, including neutral, calm, happy, sad, angry, fearful, surprise, and disgust expressions. We adopted the model structure and data pre-processing techniques outlined in (Xue et al., 2021). Intermediate features for distillation were extracted after the first linear layer for both image and audio models, having dimensionality of 1024. In our experiments, the audio model served as the teacher, while the image model was designated as the student. We trained baseline models using cross-entropy loss to compare predicted classes with the ground truth. The training was conducted over 100 epochs with a batch size of 64, using an SGD optimizer with momentum set at 0.9 and a learning rate of 0.01. For distillation, we maintained the same training options (such as epoch count, etc.) and trained the models using equal weight for the task loss and the feature distillation (FD) loss.

## D.3    MM-IMDB

Multimodal IMDB dataset (MM-IMDB) (Arevalo et al., 2017) comprises diverse metadata for 25,959 movies, including their poster (image), plot (text), and genre information. The goals is to predict multiple genres when provided with either a poster or a plot. Our experimental setting follows the data split and learning strategy outlined in (Xue et al., 2021). To enable a clear observation of the effects of distillation, we utilized a downscaled version of the model provided by (Xue et al., 2021) for training, resulting in intermediate features with a dimensionality of 64. Specifically, for both the image and text encoders, we modified the linear layer to extract 64-dimensional features, consequently reducing the dimensionality of the intermediate features in the head network to a maximum of 64. The fusion network concatenates features from both the image and text encoders, followed by projecting them to a 64-dimensional space using a linear layer. We utilized both the image model and the image-text fusion model as teachers, while the text model is utilized as the student. The performance of model is evaluated using F1 score, which is calculated by the harmonic mean of precision and recall. The macro F1 score provides an average across classes, while the micro F1 score provides an average across instances. We trained baseline models with Binary Cross-Entropy loss between predicted classes and multi-label ground truth. The training was conducted over 100 epochs with a batch size of 128, using an AdamW optimizer with a learning rate of 0.001 and a weight decay of 0.01. During distillation, we maintained the same training options (such as epoch count, etc.) and trained the models using an equal weight for the task loss and the feature distillation (FD) loss.

## D.4    NUSCENES

The nuScenes benchmark (Caesar et al., 2020) is a large-scale 3D object detection dataset comprising 1,000 driving scenes. For each scene, it provides RGB images captured by six cameras covering all directions, and point clouds obtained from a LiDAR sensor. The LiDAR and camera sensors were respectively attributed as the teacher and student modalities. Despite the different backbone structures between the LiDAR teacher and image student models, both models produce aligned bird's-eye-view (BEV) features. Therefore, we utilized BEV for feature distillation. The BEV feature grid was configured as 128 (width) $\times$ 128 (height), with each grid containing a 256-dimensional feature. Consequently, BEV feature having shape of $(128, 128, 256)$, was utilized for distillation. Model performance was assessed using via nuScenes Detection Score (NDS) for 10 object classes,

where NDS considers both the mean average precision and the measurement errors for orientation, translation, velocity, scale, and attributes. We evaluated the model on the validation set. Following prior work on cross-modal distillation for 3D object detection (Chen et al., 2023), We used BEV-Former (Li et al., 2022e) with ResNet-50 image backbone as the student model (image-modality) and Object-DGCNN (Wang & Solomon, 2021) as the teacher model (LiDAR-modality). Similar to (Li et al., 2022e), the baseline student models were trained for 24 epochs using a learning rate of $2 \times 10^{-4}$ and a batch size of 1 per GPU. We employed AdamW as the optimizer with a weight decay of $1 \times 10^{-2}$. Following the hyper-parameters provided by (Chen et al., 2023; Wang & Solomon, 2021), we pretrained teacher model for 20 epochs using a initial learning rate of $10^{-4}$ and gradually increased to $10^{-3}$ which is finally decreased to $10^{-8}$. For distillation, we maintained the same training hyper-parameters for the student while freezing the weights of the pretrained teacher. To solely evaluate the effect of the CIBA framework, we did not use any training tricks or test-time augmentation in our experiments. All models were trained on 8 of NVIDIA A100 GPU while following the original codebase from (Chen et al., 2023; Li et al., 2022e; Wang & Solomon, 2021).

### D.5 VGG-SOUND

VGG-Sound (Chen et al., 2020) is a large-scale video event classification dataset comprising over 210,000 video clips across 310 audio classes. Each clip is approximately 10 seconds long, resulting in more than 550 hours of audio-visual data. For our experiments, we utilized a subset of 100 classes, constructing training, validation, and test sets with 50,000, 5,000, and 5,000 samples, respectively, following the setup described in Pian et al. (2023). We adopted the model structure and data pre-processing techniques outlined in (Xue et al., 2021). The video frames were resized to a resolution of $(256 \times 256)$. Audio data was transformed into 2-dimensional spectrograms using the short-time Fourier transform, with the following parameters: window size = 1024, hop size = 512, and sampling rate = 16,000. Each spectrogram was then resized to $(513 \times 313)$ and provided as input to the network. For the backbone model, we employed ResNet (He et al., 2016) for both audio and video modalities. Since the audio spectrogram is a single-channel image, the input channel of the first layer in the audio ResNet was modified to 1. Features extracted after the average pooling layer were utilized for the distillation. We trained the baseline models using cross-entropy as the task loss. The training process spanned 50 epochs with a batch size of 32, utilizing the AdamW optimizer with a weight decay of 0.00005 and a learning rate of 0.001. For the distillation experiments, the same training settings (e.g., epoch count and batch size) were applied. The models were trained with equal weighting for the task loss and the feature distillation (FD) loss, ensuring a balanced contribution from both objectives during optimization.

### D.6 DETAILS FOR FIGURE 1

The specific process for creating Fig.1 is as follows: First, we extract features from the training data for each of the four models presented in Fig.1: (a) audio baseline, (b) image baseline (w/o distillation), (c) image model trained with MSE distillation, and (d) image model trained with our CIBA framework. The extracted features form matrices of size (feature dimension $D$) by (number of samples $N$). Then, all features are concatenated along the dimension axis to form a $4D \times N$ matrix, which is subsequently projected into a 2D space using the t-SNE algorithm (i.e., $4D \times N \to 4D \times 2$). It should be noted that the projection is performed along $N$, not $D$, to observe the distribution of modality-general and modality-specific information inherent in the learned features. Finally, for clearer comparisons of the projected features, we present visualizations of each image model's features alongside the teacher (audio) model's features.

# E   ADDITIONAL EXPERIMENTS AND ANALYSIS

## E.1   STATISTICAL ANALYSES OF TABLE 1

We provided statistical analyses of the results from Tab.1, including significance probability (p-value in Tab.4) and box plots (Fig.7). The results in Tab.4 and Fig.7 demonstrate that in most cases, the proposed framework achieves a statistically significant (i.e., p-value $< 0.05$ or box distributions exhibit significant differences) improvement in performance, compared to the MSE and other baseline. In the fusion to text scenario of the MM-IMDB dataset, we confirmed that the F1-macro performance is significantly improved compared to MSE. The MM-IMDB is a long-tailed dataset (Arevalo et al., 2017), where the largest class has approximately 40 times more data than the smallest class. Therefore, the performance improvement in F1-macro, which measures the average performance across classes, indicates that our proposed method enables the student model to learn diverse and discriminative features, highlighting the validity of this result.

Table 4: Statistical analyses of experimental results of Tab.1. Modality 'A', 'I', and 'T' denote Audio, Image, and Text, respectively. The number within the parentheses denotes the dimension of the bottleneck feature. All experiments were repeated five times with random seeds, and we report the mean and standard deviation of results as $mean_{std}$. Additionally, for the statistical significance test, we present the p-value for the performance difference between implementing the proposed CIBA framework and not implementing it. '$< 0.001$' indicates a very small value that is less than 0.001. Generally, a p-value $\leq 0.05$ indicates a significant difference.

(a) RAVDESS dataset

| Method | Modality | Val. | p-value | Test | p-value |
|---|---|---|---|---|---|
| Audio-baseline | A | $73.26_{1.51}$ | | $72.22_{2.26}$ | |
| Image-baseline | I | $81.64_{0.79}$ | | $78.08_{1.30}$ | |
| MSE | $A \rightarrow I$ | $80.91_{0.68}$ | | $78.42_{0.69}$ | |
| MSE + CIBA (8) | $A \rightarrow I$ | $83.74_{0.98}$ | 0.001 | $81.64_{1.38}$ | 0.004 |
| CE | $A \rightarrow I$ | $82.17_{0.70}$ | | $78.83_{0.93}$ | |
| CE + CIBA (8) | $A \rightarrow I$ | $83.22_{0.51}$ | 0.028 | $81.19_{1.58}$ | 0.026 |
| CLIP | $A \rightarrow I$ | $81.33_{1.23}$ | | $78.79_{1.53}$ | |
| CLIP + CIBA (64) | $A \rightarrow I$ | $83.80_{0.39}$ | 0.009 | $80.77_{1.04}$ | 0.048 |
| Margin | $A \rightarrow I$ | $81.82_{0.81}$ | | $78.62_{1.24}$ | |
| Margin + CIBA (128) | $A \rightarrow I$ | $83.46_{0.94}$ | 0.019 | $80.42_{1.14}$ | 0.044 |

(b) MM-IMDB datatset

| Method | Modality | F1-micro | p-value | F1-macro | p-value |
|---|---|---|---|---|---|
| Image-baseline | I | $40.00_{0.50}$ | | $25.82_{0.63}$ | |
| Text-baseline | T | $57.87_{0.27}$ | | $45.95_{0.38}$ | |
| Fusion-baseline | F (I+T) | $57.09_{0.41}$ | | $46.10_{0.41}$ | |
| MSE | $I \rightarrow T$ | $56.74_{0.22}$ | | $42.18_{0.48}$ | |
| +CIBA w/ DVIB (16) | $I \rightarrow T$ | $59.29_{0.16}$ | $< 0.001$ | $47.71_{0.22}$ | $< 0.001$ |
| +CIBA w/ SA (16) | $I \rightarrow T$ | $58.86_{0.54}$ | $< 0.001$ | $46.51_{0.66}$ | $< 0.001$ |
| +CIBA w/ VQ-VAE (16) | $I \rightarrow T$ | $58.67_{0.45}$ | $< 0.001$ | $46.71_{0.37}$ | $< 0.001$ |
| MSE | $F \rightarrow T$ | $58.32_{0.28}$ | | $45.77_{0.16}$ | |
| +CIBA w/ DVIB (32) | $F \rightarrow T$ | $58.73_{0.38}$ | 0.092 | $47.11_{0.46}$ | 0.002 |
| +CIBA w/ SA (32) | $F \rightarrow T$ | $58.46_{0.16}$ | 0.370 | $46.19_{0.34}$ | 0.051 |
| +CIBA w/ VQ-VAE (32) | $F \rightarrow T$ | $58.48_{0.18}$ | 0.340 | $46.44_{0.33}$ | 0.002 |

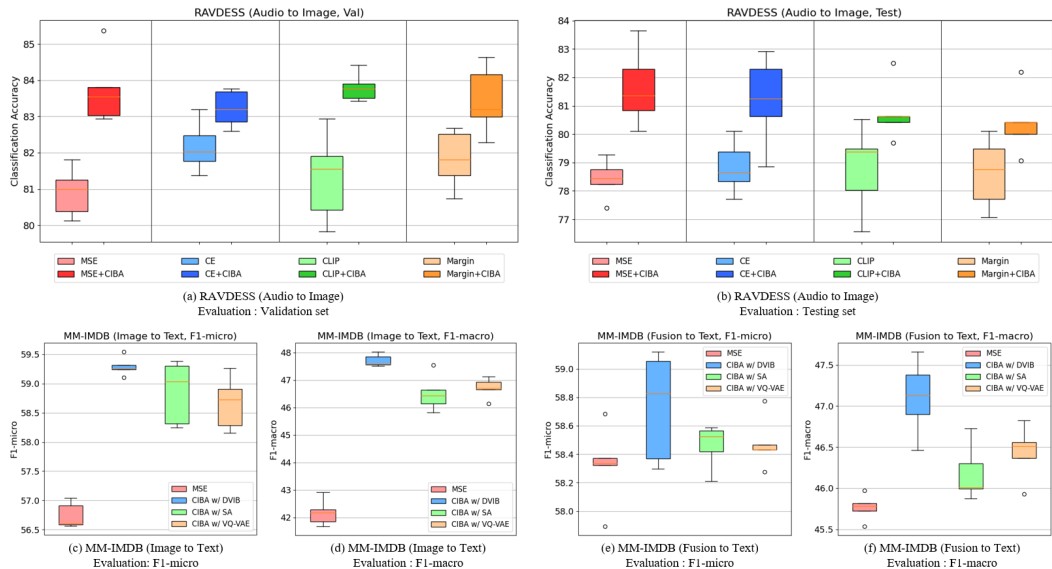

Figure 7: Statistical validation of the experimental results in Table 1. All experiments were repeated five times with random seeds, and box plots of the results are provided. **RAVDESS (a-b):** The darkly shaded box represents the results from applying our CIBA framework, while the lightly shaded box represents the results without its application. The y-axis indicates classification accuracy. **MM-IMDB (c-f):** (c-d) correspond to the results for the fusion to text distillation scenario, while (e-f) correspond to the results for the image to text distillation scenario. The y-axis represents the F1-score, with (c,e) reporting the F1-micro score and (d,f) reporting the F1-macro score. The red box represents the performance of MSE baseline, and the other colors represent the performance when the proposed framework is applied.

## E.2 STATISTICAL ANALYSES OF TABLE 3

We provided statistical analyses of the results from Tab.3, including significance probability (p-value in Tab.5). The results in Tab.5 demonstrate that in all cases, the proposed framework achieves a statistically significant (i.e., p-value $< 0.05$) improvement in performance, compared to the MSE.

Table 5: Statistical analyses of experimental results on VGG-Sound dataset. Modality 'V' and 'A' denote Video and Audio, respectively. Subscript '50' and '18' under the modality symbol indicate that the ResNet-50 and ResNet-18 models are employed as backbone models. The number within the parentheses denotes the dimension of the bottleneck feature. All experiments were repeated five times with random seeds, and we report the mean and standard deviation of results as $mean_{std}$. Additionally, for the statistical significance test, we present the p-value for the performance difference between implementing the proposed CIBA framework and not implementing it. '$< 0.001$' indicates a very small value that is less than 0.001. Generally, a p-value $\leq 0.05$ indicates a significant difference.

Table 6: VGG-Sound dataset

| Method | Modality | Val. | p-value | Test | p-value |
|---|---|---|---|---|---|
| Video-baseline (ResNet-50) | $V_{50}$ | $50.42_{0.37}$ | | $49.43_{0.66}$ | |
| Audio-baseline (ResNet-50) | $A_{50}$ | $69.55_{0.36}$ | | $68.76_{0.33}$ | |
| Video-baseline (ResNet-18) | $V_{18}$ | $42.11_{0.53}$ | | $41.53_{0.41}$ | |
| Audio-baseline (ResNet-18) | $A_{18}$ | $68.86_{0.33}$ | | $69.08_{0.52}$ | |
| MSE | $V_{18} \rightarrow A_{18}$ | $67.54_{0.76}$ | | $68.32_{0.49}$ | |
| MSE + CIBA (16) | $V_{18} \rightarrow A_{18}$ | $70.11_{0.40}$ | $< 0.001$ | $70.39_{0.48}$ | $< 0.001$ |
| MSE | $A_{18} \rightarrow V_{18}$ | $42.61_{0.33}$ | | $41.28_{0.57}$ | |
| MSE + CIBA (16) | $A_{18} \rightarrow V_{18}$ | $43.59_{0.57}$ | $0.015$ | $42.55_{0.62}$ | $< 0.001$ |
| MSE | $V_{50} \rightarrow A_{18}$ | $68.53_{0.36}$ | | $68.54_{0.34}$ | |
| MSE + CIBA (16) | $V_{50} \rightarrow A_{18}$ | $70.21_{0.28}$ | $< 0.001$ | $70.71_{0.52}$ | $< 0.001$ |
| MSE | $A_{50} \rightarrow V_{18}$ | $41.40_{0.71}$ | | $40.33_{0.39}$ | |
| MSE + CIBA (16) | $A_{50} \rightarrow V_{18}$ | $43.44_{0.33}$ | $0.001$ | $42.95_{0.19}$ | $< 0.001$ |

### E.3 VARIOUS DISTILLATION SCENARIOS ON RAVDESS DATASET

We conducted additional experiments to explore the feasibility of the proposed CIBA framework for various distillation scenarios, including image-to-audio and fusion-to-image distillation. As shown in Tab.7, CIBA also performs effectively in such settings. Notably, similar to the results in Tab.1 (b), MSE shows only marginal performance improvements over the text-baseline for the fusion teacher, while CIBA demonstrates significant performance enhancement.

Table 7: Results on the RAVDESS. Modality 'A' and 'I' denote Audio and Image, respectively. The number within the parentheses denotes the dimension of the bottleneck feature. All experiments were repeated five times with random seeds, and we report the mean and standard deviation of results as $mean_{std}$. Additionally, for the statistical significance test, we present the p-value for the performance difference between implementing the proposed CIBA framework and not implementing it. '< 0.001' indicates a very small value that is less than 0.001. Generally, a p-value $\leq 0.05$ indicates a significant difference.

| Method | Modality | Val. | p-value | Test | p-value |
|---|---|---|---|---|---|
| Audio-baseline | A | $73.26_{1.51}$ | | $72.22_{2.26}$ | |
| Image-baseline | I | $81.64_{0.79}$ | | $78.08_{1.30}$ | |
| Fusion-baseline | F (I+A) | $87.17_{0.94}$ | | $87.31_{1.46}$ | |
| MSE | A $\rightarrow$ I | $80.91_{0.68}$ | | $78.42_{0.69}$ | |
| MSE + CIBA (8) | A $\rightarrow$ I | $83.74_{0.98}$ | 0.001 | $81.64_{1.38}$ | 0.004 |
| MSE | F $\rightarrow$ I | $81.12_{0.23}$ | | $78.60_{0.83}$ | |
| MSE + CIBA (8) | F $\rightarrow$ I | $82.88_{0.56}$ | 0.001 | $80.23_{0.95}$ | 0.021 |
| MSE | I $\rightarrow$ A | $71.92_{1.35}$ | | $67.61_{1.82}$ | |
| MSE + CIBA (2) | I $\rightarrow$ A | $77.00_{1.48}$ | < 0.001 | $76.19_{3.03}$ | 0.001 |

### E.4 COMPLEXITY OF CIBA FRAMWORK

Our CIBA framework extracts and transfers modality-general components through an additional information bottleneck module for effective CMFD as described in Sec.4. Although this additional pre-training phase may seem to increase the training complexity, it only requires a small amount of computational resources since the bottleneck model typically has fewer parameters compared to the student and teacher models. For example, in the audio-image setting (Sec.5.1), the bottleneck model has more than six times fewer parameters compared to the student model (2.1M vs. 13.1M). Moreover, the proposed framework significantly improves the performance of the student model without increasing inference complexity since the bottleneck model is not required at test time. Hence, CIBA framework offers practically significant and efficient CMFD method in real-world scenarios.

### E.5 EXTENSION OF CLAIM 1 TO OTHER DISTILLATION LOSS

Optimal weights are typically determined through the gradient of a loss function with respect to the weights. First, let us compare the gradients of the MSE and Cross-entropy (CE) losses with respect to the weight. Following the notation in Lemma 3 in Appendix A, the features for the $i$-th sample $\mathbf{x}'_i$, $\mathbf{x}_i$ of the teacher and student can be represented as $\mathbf{z}'_i$, $\mathbf{z}_i$, respectively. Then, in order to calculate the cross-entropy between features, each feature can be transformed into a probability value $\mathbf{p}'_i$, $\mathbf{p}_i$ by the softmax function. The derivative of the CE loss ($-\mathbf{p}'_i \log(\mathbf{p}_i)$) with respect to $\mathbf{z}_i$ can be calculated as $\mathbf{p}_i - \mathbf{p}'_i$. On the other hand, the derivative of the MSE loss with respect to $\mathbf{z}_i$ is calculated as $\mathbf{z}_i \grave{} \mathbf{z}'_i$, as shown in Eq.11. Although the softmax function normalizes the scale of input values, it should not significantly alter the distribution itself. Therefore, we can infer that the results with CE loss would be similar to those observed with MSE loss.

To further validate such insights, we conducted an additional experiment by applying cross-entropy (CE) instead of MSE as the FD loss on the synthetic dataset settings of Sec.3.5. Fig.8 presents the expanded experimental results applying MSE, from Fig.3 in Sec.3.5. Fig.9 shows the results of the same experiment conducted with CE loss, and we found that the model still suffers from dimensional collapse. Additionally, the results in Tab.1 of the main text demonstrate that performance improvements were achieved when the proposed CIBA framework is combined with various distillation losses. These results support that our claims can be extended to other distillation losses.

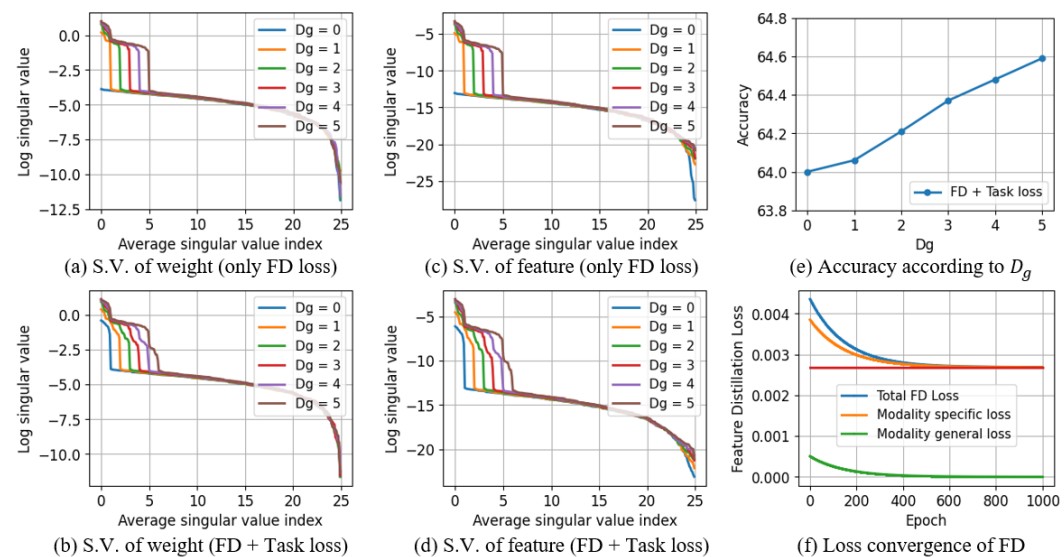

Figure 8: Experimental results on the synthetic dataset where MSE was applied as a feature distillation loss. **(a)**-**(d)** The log singular value spectrum of student's weights and features learned through distillation, according to the modality general dimension $D_g$. It should be noted that dimensional collapse still occurs when using CE loss. **(e)** The performance of the student model depending on $D_g$. **(f)** The distillation loss dynamics during training, separated into modality-general and modality-specific components. The horizontal red line represents the values derived from Corollary 1.1.

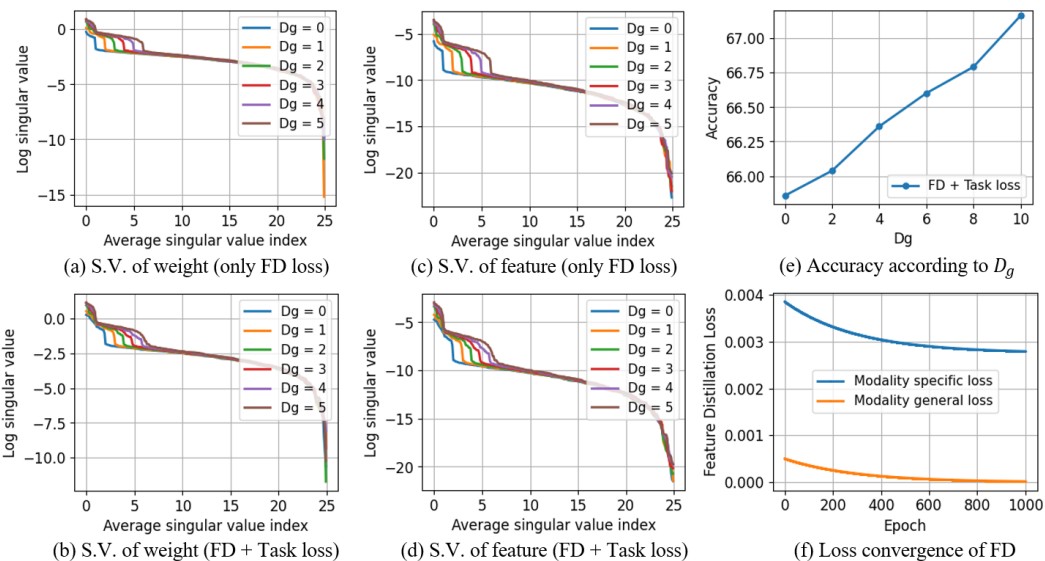

Figure 9: Experimental results on the synthetic dataset where Cross-entropy (CE) was applied as a feature distillation loss. **(a)**-**(d)** The log singular value spectrum of student's weights and features learned through distillation, according to the modality general dimension $D_g$. It should be noted that dimensional collapse still occurs when using CE loss. **(e)** The performance of the student model depending on $D_g$. **(f)** The distillation loss dynamics during training, separated into modality-general and modality-specific components. The modality general loss asymptotically converges to zero, while modality-specific loss does not. This result implies the importance of transferring only modality-general information in CMFD.

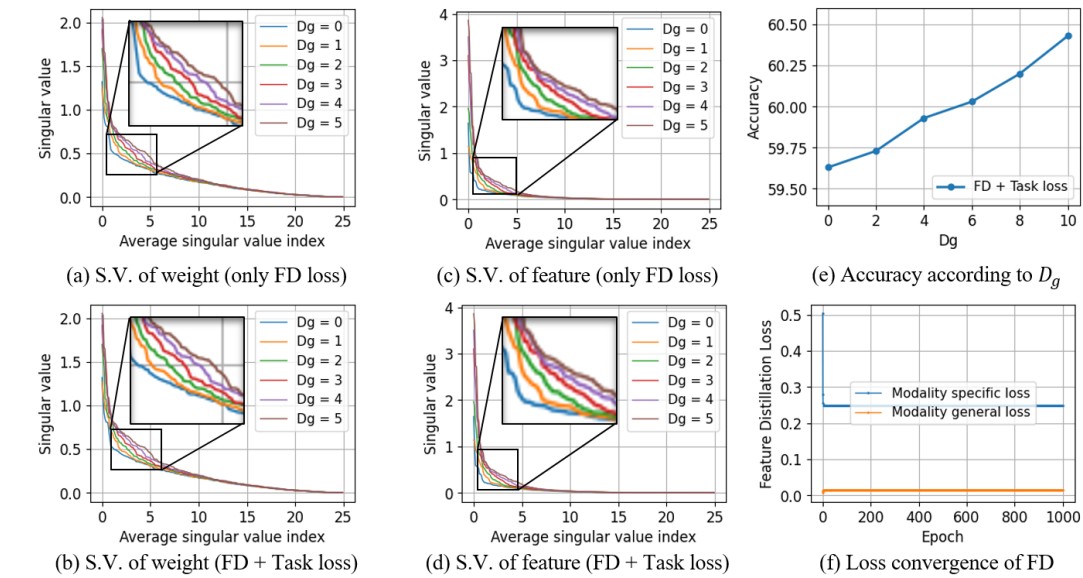

Figure 10: Experimental results on the synthetic dataset where the orthogonality assumption is relaxed by removing Gram-Schmidt process when generating synthetic data. **(a)**-**(d)** The log singular value spectrum of student's weights and features learned through distillation, according to the modality general dimension $D_g$. **(e)** The performance of the student model depending on $D_g$. **(f)** The distillation loss dynamics during training, separated into modality-general and modality-specific components.

### E.6 RELAXATION OF ORTHOGONALITY ASSUMPTION

We conducted additional experiments by relaxing the orthogonality assumption from the synthetic datasets described in Sec.3.5. Specifically, we removed the Gram-Schmidt process during synthetic data generation, meaning the data were generated following a unit normal Gaussian distribution and were not fully orthogonal. Other than this modification, we retained the same settings as outlined in Appendix D.1 and performed identical experiments in Fig.8.

The results presented in Fig.10, show that the spectrum of singular values for the student features becomes broader as the dimension of modality-general features increases, while the distributions are less distinct compared to those derived from orthogonal features in Fig.8 due to the relaxed condition. Additionally, as shown in Fig.10-(c), the trend of performance improvement with an increasing dimension of modality-general features also remains consistent. These findings confirm that our claim remains valid even when the orthogonality assumption is relaxed.

### E.7 ADDITIONAL ABLATION STUDY ON BOTTLENECK DIMENSION $H$

First, we illustrate the performance variations on the MM-IMDB dataset depending on $H$ in Fig. 11. The black line represents the performance of the MSE-based approach. Consistent with the analysis in Sec. 5.1.3, the performance tends to degrade for excessively small or large $H$ values. However, apart from these extreme $H$ values, the proposed method generally outperforms the MSE method.

In Fig.12, we also plotted the performance variations (lower plot) and the singular value spectrum of the bottleneck model's output (upper plot) with respect to $H$ on the VGG-Sound dataset, similar to Fig.5-(b) and -(c). Consistent with previous results, the proposed method significantly outperforms the MSE approach, except for extreme $H$ values.

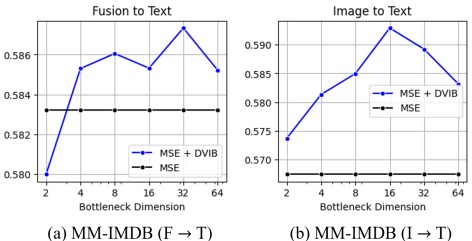

Figure 11: Performance trends on the MM-IMDB dataset with varying bottleneck dimensions $H$.

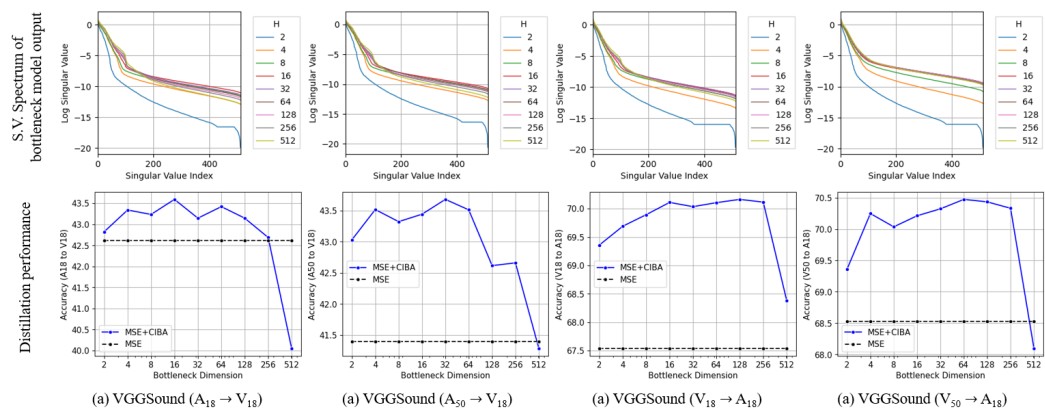

(a) VGGSound ($A_{18} \to V_{18}$)  (a) VGGSound ($A_{50} \to V_{18}$)  (a) VGGSound ($V_{18} \to A_{18}$)  (a) VGGSound ($V_{50} \to A_{18}$)

Figure 12: Experimental results on VGG-Sound. **(Upper plot)** The log singular value spectrum of the bottleneck model's output depending on the dimension of bottleneck feature $H$. **(Lower plot)** The performance trend of CIBA implemented with the sub-dimensional distillation using MSE, depending on $H$. Black line denotes the performance of MSE baseline.

Furthermore, the method described in Sec.5.1.3 for estimating an adequate $H$ value based on the singular value spectrum analysis of the bottleneck model's output features can also be applied to the VGG-Sound dataset. In the upper plot of Fig.12 of Appendix E.7, the singular value spectrum saturates around $H = 16$ across all scenarios. Although the lower plot of Fig.12 shows that $H = 16$ does not always achieve the optimal performance, it consistently delivers sufficiently strong results. These findings demonstrate that the proposed method for selecting $H$ is practical and applicable to real-world scenarios.

