# OpenReview forum: "Understanding Dimensional Collapse in Cross-Modal Feature Distillation"
_ICLR.cc/2025/Conference — Submitted to ICLR 2025_

### Official Review · Reviewer_i91g · 2024-10-27

**Soundness:** 3
**Presentation:** 3
**Contribution:** 2
**Rating:** 6
**Confidence:** 3

**Summary:**

This paper investigates the problem of dimensional collapse in cross-modal feature distillation (CMFD), where a student model trained on one modality aims to mimic the feature representations of a teacher model trained on a different modality. The authors hypothesize that the distributional shift, or "modality gap", between the teacher and student modalities leads to the student's feature space collapsing to only capture the modality-general features, resulting in suboptimal distillation performance. To address this issue, the authors provide in-depth analysis on how distributional shifts across different modalities and propose a Cross-modal Information Bottleneck Approximation (CIBA) scheme that extracts and transfers the modality-general features from the teacher to the student, allowing the student to effectively span both modality-general and modality-specific feature spaces.

**Strengths:**

1. Research about cross-modal knowledge distillation (CMKD) on the feature view is an important topic for multimodal learning and knowledge distillation. This paper analyses the dimensional collapse induced by modality gap and propose Cross-modal Information Bottleneck Approximation (CIBA) to disentangle the general and specific knowledge, which is novel and practical.
2. Utilizing the Mean Squared Error (MSE) loss for feature distillation (FD) is reasonable and suitable for the subsequent theoretical analysis.
3. This work is a good extension of the modality focusing hypothesis, and gives a solid analysis and detailed solutions.
4. This work is well written and organized. Extensive experiments on Audio-Image, Image-Text, and LiDAR-Camera crossmodal transfer are conducted.

**Weaknesses:**

1. Modality gap is widely studies in multimodal learning, and this paper does not give a review of previous modality gap analysis. Moreover, the cross-modal knowledge distillation on logit-level method [r1] is not mentioned and analysed.
[r1] Huo F, Xu W, Guo J, et al. C2KD: Bridging the Modality Gap for Cross-Modal Knowledge Distillation[C]//Proceedings of the IEEE/CVF Conference on Computer Vision and Pattern Recognition. 2024: 16006-16015.
2. The RAVDESS, MM-IMDB, and nuScenes have limited class categories. Large-scale experiments like conducting experiments on VGG-Sound (or subset) will make the paper more convincing.
3. Related works about the 'Cross-modal knowledge distillation' are somewhat out-of-date, only one paper published in 2023 is mentioned.
4. The proposed method is somewhat similar to online distillation [r1] and task-oriented feature distillation [r2]. How about the performance of directly employing task-oriented feature distillation [r2] on cross-modal feature distillation?
[r2]Zhang L, Shi Y, Shi Z, et al. Task-oriented feature distillation[J]. Advances in Neural Information Processing Systems, 2020, 33: 14759-14771.

**Questions:**

1. How is the Figure 1 formulated? The manuscript does not mention the details. I think it is important for the motivation of modality-general and modality-specific knowledge analysis.
2. How about directly apply unimodal knowledge distillation on crossmodal knowledge distillation? Could the proposed method be integrated into SOTA methds?

[r1] Zhang L, Shi Y, Shi Z, et al. Task-oriented feature distillation[J]. Advances in Neural Information Processing Systems, 2020, 33: 14759-14771.

[r2] Huang T, You S, Wang F, et al. Knowledge distillation from a stronger teacher[J]. Advances in Neural Information Processing Systems, 2022, 35: 33716-33727.

---

> ### Author Response · Authors · 2024-11-22
>
> We sincerely appreciate your positive evaluation of our work. In particular, we are grateful for your recognition of **the significance of our contributions to CMKD literature and the accompanying modality gap analysis**, as well as **the novelty of the proposed CIBA framework**. Additionally, we thank you for highlighting our **theoretical contributions**, **extensive experiments**, and **writing quality**.
>
> &nbsp;
>
> ## **Additional Experimental Validation on VGG-Sound dataset**
>
> To address the reviewer’s concerns, we conducted **additional experiments on VGG-Sound dataset**. For detailed experimental results and analyses, please refer to Section 5.4 and Appendix E.2 of the revised document. For your convenience, we provide a summary of the key findings in the table below.
> Here we observe that **our proposed method achieves significant performance improvements compared to the MSE baseline in both video-to-audio and audio-to-video scenarios**. Furthermore, we conducted experiments using various backbones and consistently observed performance gains.
>
> >**VGG-Sound**
> | Method | Modality | Val. | Test |
> |---|:---:|:---:|:---:|
> | Video (ResNet50 backbone) | V50 | 50.42 | 49.43 |
> | Audio (ResNet50 backbone) | A50 | 69.55 | 68.76 |
> | Video (ResNet18 backbone) | V18 | 42.11 | 41.53 |
> | Audio (ResNet18 backbone) | A18 | 68.86 | 69.08 |
> |||||
> | MSE         | V18 $\rightarrow$ A18 | 67.54 | 68.32 |
> | MSE + CIBA  | V18 $\rightarrow$ A18 | 70.11 | 70.39 |
> |||||
> | MSE         | V50 $\rightarrow$ A18 | 68.53 | 68.54 |
> | MSE + CIBA  | V50 $\rightarrow$ A18 | 70.21 | 70.71 |
> |||||
> | MSE         | A18 $\rightarrow$ V18 | 42.61 | 41.28 |
> | MSE + CIBA  | A18 $\rightarrow$ V18 | 43.59 | 42.55 |
> |||||
> | MSE         | A50 $\rightarrow$ V18 | 41.40 | 40.33 |
> | MSE + CIBA  | A50 $\rightarrow$ V18 | 43.44 | 42.95 |
>
>
> &nbsp;
>
> ## **Comparison to Task-oriented Feature Distillation**
>
> Task-oriented feature distillation (TOFD) is a methodology that extracts "task-oriented information" from the teacher's features by simultaneously training an auxiliary classifier during the distillation process. Following the reviewer’s suggestion, we directly **applied TOFD to the VGG-Sound dataset**, and the experimental results are presented in the table below.
>
> >**VGG-Sound**
> | Method | Modality | Val. | Test |
> |---|:---:|:---:|:---:|
> | Video (ResNet18 backbone) | V18 | 42.11 | 41.53 |
> | Audio (ResNet18 backbone) | A18 | 68.86 | 69.08 |
> |||||
> | MSE            | V18 $\rightarrow$ A18 | 67.54 | 68.32 |
> | Task-Oriented  | V18 $\rightarrow$ A18 | 67.23 | 67.73 |
> | MSE + CIBA     | V18 $\rightarrow$ A18 | 70.11 | 70.39 |
>
> To the best of our understanding, **TOFD does not explicitly address the separation of modality-general and modality-specific features**, which is crucial for improving the quality of cross-modal distillation. As a result, TOFD, like MSE, exhibits suboptimal performance due to the influence of modality gaps.
>
>
> &nbsp;
>
> ## **Integrating Uni-Modal Feature Distillation Methods into Our Framework**
>
> In uni-modal feature distillation settings, there is **no need to consider modality-general and modality-specific features, as the modality of inputs for the teacher and student models are identical**. As a result, global feature distillation methods--which force the student model's features to mimic the whole features of the teacher--are typically employed and have demonstrated strong performance.
> Although MSE and cross-entropy (CE) losses have contributed to improving the quality of uni-modal knowledge distillation, these approaches are less effective in cross-modal settings due to the modality gap, which hinders effective cross-modal distillation, as demonstrated in Tab.1, 2, and 3.
> In the above comment, we also have shown that task-oriented uni-modal distillation approach lead to suboptimal cross-modal distillation results.
> Instead, our CIBA framework achieves improved quality of cross-modal distillation.

---

> ### Author Response · Authors · 2024-11-22
>
> ## **Additional Related Works**
>
> As we mentioned in the introduction (L45-L49), our paper **primarily explores how the modality gap in cross-modal settings can lead to dimensional collapse in the student model**. Consequently, our main focus is on addressing cross-modal **"feature"** distillation, where dimensional collapse can severely degrade the distillation quality.
> At the same time, we acknowledge the importance of logit-level (output-level) knowledge distillation, as both logit- and feature-level KD can offer complementary insights to synergistically address the modality gap problem. This represents an intriguing direction for future work, as discussed in Section 6 (Discussion and Conclusion).
> In response to the reviewers' feedback, we have introduced additional concurrent literature and drawn comparisons in the context of the modality gap in Section 2.1 of the revised document.
>
> &nbsp;
>
> ## **Details of Figure 1**
>
> As you mentioned, **Fig.1 effectively illustrates the motivation behind our work**. In response to your feedback, we have included additional details for Fig.1 in Appendix D.6 of the revised document.
> To summarize, the process for creating Fig.1 is as follows:
> First, we extract features from the training data for each of the four models presented in Fig.1: (a) audio baseline, (b) image baseline (w/o distillation), (c) image model trained with MSE distillation, and (d) image model trained with our CIBA framework. The extracted features form matrices of size $D$ (feature dimension) by $N$ (number of samples).
> Then, all features are concatenated along the dimensional axis to form a $4D \times N$ matrix, which is subsequently projected into a 2D space using the t-SNE algorithm (i.e., $4D\times N \rightarrow 4D \times 2$). Please note that the projection is performed along $N$, not $D$, to observe the distribution of modality-general and modality-specific information inherent in the learned features.
> Finally, to enable clear comparisons of the projected features, we present visualizations of each image model’s features alongside those of the teacher (audio) model.

---

> > ### Comment · Reviewer_i91g · 2024-11-23
> > **Official Comment by Reviewer i91g**
> >
> > Thank you for the detailed responses and additional experiments! I maintain the original score.

---

> > > ### Author Response · Authors · 2024-11-26
> > >
> > > We sincerely appreciate your recognition of the contributions of our work and your decision to maintain the acceptance score. Thank you again for your careful consideration and the time and effort you have dedicated to this review.

---

### Official Review · Reviewer_nuh2 · 2024-11-02

**Soundness:** 3
**Presentation:** 3
**Contribution:** 3
**Rating:** 6
**Confidence:** 3

**Summary:**

This paper investigates the relationship between distributional shifts across modalities and their impact on the effectiveness of cross-modal knowledge distillation (CMKD), specifically addressing the issue of cross-modal feature distillation. The authors hypothesize and validate that the modality gap between the teacher and student models may lead to dimensional collapse in the student’s feature space. To address this, they propose a Cross-modal Information Bottleneck Approximation (CIBA) scheme aimed at extracting and transferring modality-general features from the teacher model. Experimental results demonstrate that the proposed distillation strategy effectively mitigates dimensional collapse in the student model.

**Strengths:**

1.	The authors successfully propose and validate that the modality gap between the teacher and student models can lead to dimensional collapse in the student’s feature space.

2.	A novel Cross-modal Information Bottleneck Approximation (CIBA) scheme is introduced to extract and transfer modality-general features from the teacher model.

3.	Experimental results across various loss functions and tasks provide strong evidence for the effectiveness of the proposed method.

**Weaknesses:**

1.	The work is predicated on the assumption of linear feature extractors; however, in practical applications, most feature extractors are non-linear.

2.	In the MM-IMDB dataset, the observed improvement is marginal. Could you please provide a more detailed explanation for this finding?

**Questions:**

See the weaknesses.

---

> ### Author Response · Authors · 2024-11-22
>
> We sincerely appreciate your positive evaluation of our work. In particular, we are grateful for your acknowledgment of the core contributions of our paper: **investigation of the influence of the modality gap on CMFD in relation to the dimensional collapse phenomenon**, as well as **proposing CIBA framework** and its **experimental validation** aimed at mitigating the impact of the modality gap.
>
> We would also be deeply grateful if you could *consider providing an improved evaluation*, should our revisions have sufficiently addressed your concerns.
>
> &nbsp;
>
> ## **Empirical Extension of Theoretical Analysis to Non-linear Settings**
>
> While providing theoretical analyses of dimensional collapse in non-linear settings may seem appealing, the pursuit of such analyses itself constitutes a distinct and interesting field of research, encompassing areas such as identifiability and independent component analysis [1,2]. It is worth noting that **prior works on the modality focusing hypothesis [3] and dimensional collapse [4] also established their theoretical validity using linear feature extractors**, likely due to the inherent challenges of addressing non-linear settings.
>
> We would like to emphasize that we have **empirically demonstrated how theoretical insights from simple linear settings can be extended to complex non-linear settings**, including real-world multi-modal datasets. Specifically, in Section 5.1.1, we showed that the MSE loss still causes dimensional collapse in real-world settings (Fig.5(a)), whereas our CIBA framework successfully alleviates this issue (Tab.1(a)).
>
> > [1] Hälvä et al., Disentangling identifiable features from noisy data with structured nonlinear ICA, NeurIPS, 2021. \
> [2] Hyvärinen et al., Nonlinear independent component analysis for principled disentanglement in unsupervised deep learning, Patterns, 2023. \
> [3] Xue et al., The modality focusing hypothesis: Towards understanding crossmodal knowledge distillation, ICLR 2024. \
> [4] Jing et al., Understanding dimensional collapse in contrastive self-supervised learning, ICLR 2022.
>
> &nbsp;
>
> ## **Significance of Experimental Results on MM-IMDB Dataset**
>
> In Appendix E.1, we provided statistical analyses of the results from Tab.1, including an analysis of the results on the MM-IMDB dataset. To summarize, the proposed method demonstrates **a statistically significant improvement in F1-macro performance** compared to the MSE baseline. Notably, on **the long-tailed MM-IMDB dataset**, the improvement in F1-macro, which measures the average performance across classes, highlights that **our proposed method enables the student model to learn diverse and discriminative features**, further validating these findings.

---

> ### Comment · Reviewer_nuh2 · 2024-11-27
> **Thanks for your comment!**
>
> Thanks for your response! I maintain the original score.

---

> > ### Author Response · Authors · 2024-11-27
> >
> > We sincerely appreciate your recognition of the contributions of our work and your decision to maintain the acceptance score. Thank you again for your careful consideration and the time and effort you have dedicated to this review.

---

### Official Review · Reviewer_AFwU · 2024-11-05

**Soundness:** 3
**Presentation:** 3
**Contribution:** 3
**Rating:** 6
**Confidence:** 3

**Summary:**

This paper tries to address the challenges associated with deploying multi-modal neural networks in real-world applications, specifically focusing on the constraints of limited computing resources and complex sensor configurations. The authors explore Cross-Modal Knowledge Distillation (CMKD) as a solution for transferring knowledge from a pretrained teacher model to a more deployable student model tailored to a target modality. Despite the advancements in CMKD across various domains, the paper identifies a gap in understanding how distributional shifts between modalities—referred to as the modality gap—affect the efficacy of feature distillation. The study hypothesizes and empirically demonstrates that a significant modality gap leads to dimensional collapse within the student model's feature space, undermining performance. To mitigate this issue, the authors introduce the Cross-modal Information Bottleneck Approximation (CIBA) scheme, designed to extract and transfer modality-general features from the teacher model effectively. Experimental results on diverse real-world multi-modal datasets confirm that the proposed CIBA method successfully reduces dimensional collapse in the student model, resulting in enhanced performance. This work contributes a deeper understanding of the interplay between modality gaps and knowledge transfer in CMKD, offering a practical solution to improve the deployment of multi-modal neural networks under resource constraints.

**Strengths:**

1. The theoretical and empirical investigation of the "modality gap"—the distributional shifts between different modalities—and its detrimental effect on CMKD, specifically leading to dimensional collapse in the student model’s feature space.

2. CIBA extracts modality-general features from the teacher model and transfers them to sub-dimensions of the student’s features. This method mitigates the dimensional collapse, ensuring more robust and effective knowledge transfer.

**Weaknesses:**

1. Since RAVDESS is a relatively small size dataset. Do you try to work on VGGSound?

**Questions:**

N/A

---

> ### Author Response · Authors · 2024-11-22
>
> We sincerely appreciate your positive evaluation of our work. In particular, we are grateful for your acknowledgment of the core contributions of our paper: **investigation of the influence of the modality gap on CMFD in relation to the dimensional collapse phenomenon**, as well as **proposing CIBA framework** and its **experimental validation** aimed at mitigating the impact of the modality gap.
>
> We would also be deeply grateful if you could *consider providing an improved evaluation*, should our revisions have sufficiently addressed your concerns.
>
> &nbsp;
>
> ## **Additional Experimental Validation on VGG-Sound dataset**
>
> To address the reviewer’s concerns, we conducted **additional experiments on VGG-Sound dataset**. For detailed experimental results and analyses, please refer to Section 5.4 and Appendix E.2 of the revised document. For your convenience, we provide a summary of the key findings in the table below.
> Here we observe that **our proposed method achieves significant performance improvements compared to the MSE baseline in both video-to-audio and audio-to-video scenarios**. Furthermore, we conducted experiments using various backbones and consistently observed performance gains.
>
> >**VGG-Sound**
> | Method | Modality | Val. | Test |
> |---|:---:|:---:|:---:|
> | Video (ResNet50 backbone) | V50 | 50.42 | 49.43 |
> | Audio (ResNet50 backbone) | A50 | 69.55 | 68.76 |
> | Video (ResNet18 backbone) | V18 | 42.11 | 41.53 |
> | Audio (ResNet18 backbone) | A18 | 68.86 | 69.08 |
> |||||
> | MSE         | V18 $\rightarrow$ A18 | 67.54 | 68.32 |
> | MSE + CIBA  | V18 $\rightarrow$ A18 | 70.11 | 70.39 |
> |||||
> | MSE         | V50 $\rightarrow$ A18 | 68.53 | 68.54 |
> | MSE + CIBA  | V50 $\rightarrow$ A18 | 70.21 | 70.71 |
> |||||
> | MSE         | A18 $\rightarrow$ V18 | 42.61 | 41.28 |
> | MSE + CIBA  | A18 $\rightarrow$ V18 | 43.59 | 42.55 |
> |||||
> | MSE         | A50 $\rightarrow$ V18 | 41.40 | 40.33 |
> | MSE + CIBA  | A50 $\rightarrow$ V18 | 43.44 | 42.95 |

---

> > ### Author Response · Authors · 2024-12-01
> > **Feedback reminder**
> >
> > Dear Reviewer AFwU,
> >
> > We sincerely appreciate the time and effort that you have dedicated to reviewing our paper. Considering the significance of the discussion phase, we would like to ask if you could spare some time to review our rebuttal responses.
> >
> > Taking into account your comments, we have conducted additional validation on a relatively large dataset (VGG-Sound). We hope that these results will help address your concerns and lead to an improved evaluation of our work.
> >
> > Best regards,
> > The authors.

---

> > > ### Comment · Reviewer_AFwU · 2024-12-02
> > > **Official Comment by Reviewer AFwU**
> > >
> > > Thank you for your comprehensive answers and the additional experiments. I will retain the original score.

---

> > > > ### Author Response · Authors · 2024-12-03
> > > >
> > > > We appreciate your recognition of our contributions and your decision to maintain the acceptance score. Thank you again for your careful consideration and the time and effort you have dedicated to this review.

---

### Official Review · Reviewer_XgGs · 2024-11-05

**Soundness:** 4
**Presentation:** 3
**Contribution:** 2
**Rating:** 5
**Confidence:** 4

**Summary:**

The paper titled "Understanding Dimensional Collapse in Cross-Modal Feature Distillation" investigates the challenges of transferring knowledge across different modalities in multi-modal neural networks, specifically focusing on the problem of dimensional collapse in cross-modal feature distillation (CMFD). The authors hypothesize that the modality gap between the teacher and student models leads to dimensional collapse in the student's feature space, which degrades the quality of knowledge distillation. To address this, they propose a novel framework called Cross-modal Information Bottleneck Approximation (CIBA), which aims to extract and transfer modality-general features from the teacher model to sub-dimensions of the student model's features. The paper empirically demonstrates that CIBA effectively reduces dimensional collapse and improves performance on various real-world multi-modal datasets, including RAVDESS (Audio-Image), MM-IMDB (Image-Text), and nuScenes (LiDAR-Camera). The key contributions of the paper are the theoretical and empirical investigation of the modality gap's impact on CMKD, the proposal of the CIBA framework, and the validation of its effectiveness across different modalities.

**Strengths:**

The paper proposes a novel cross-modal knowledge distillation (CMKD) method by focusing on the issue of dimensional collapse in cross-modal feature distillation. The concept of modality gap and its impact on the efficacy of feature distillation is a fresh approach to understanding the limitations of CMKD. The proposal of the Cross-modal Information Bottleneck Approximation (CIBA) scheme is creative and addresses a significant problem in transferring knowledge across different modalities.

The paper is well-written. The figures and tables are clear and effectively support the textual content.

**Weaknesses:**

The contribution is incremental.

I feel the information bottleneck approximation idea has been used extensively.

While the proposed method is shown to outperform baseline approaches, it is unclear how it compares to the most recent and advanced techniques in the field, i.e., DML[1], DKD[2], DIST[3], C2KD[4].

[1] Ying Zhang, Tao Xiang, Timothy M. Hospedales, and Huchuan Lu. Deep mutual learning. In CVPR, 2018. 1, 3, 6, 7.

[2] Borui Zhao, Quan Cui, Renjie Song, Yiyu Qiu, and Jiajun Liang. Decoupled knowledge distillation. In CVPR, 2022. 1, 2, 3, 4, 6, 7.

[3] Tao Huang, Shan You, Fei Wang, Chen Qian, and Chang Xu. Knowledge distillation from a stronger teacher. In NeurIPS, 2022. 1, 2, 6, 7, 8.

[4] Huo F, Xu W, Guo J, et al. C2KD: Bridging the Modality Gap for Cross-Modal Knowledge Distillation[C]//Proceedings of the IEEE/CVF Conference on Computer Vision and Pattern Recognition. 2024: 16006-16015.

**Questions:**

How does the performance of CIBA compare when the assumption of orthogonality between modality-general and modality-specific features is relaxed?

How does CIBA differ from the previous CMKD approaches?

How sensitive is the CIBA framework to the choice of hyperparameters, particularly the dimension of the bottleneck feature (H)?

---

> ### Author Response · Authors · 2024-11-22
>
> We appreciate the constructive comments from reviewer "XgGs," recognizing the strengths of our work as **fresh approach to understanding the limitations of CMKD**, **proposing a creative distillation scheme**, and **writing quality**.
>
> We hope that our comments below adequately address your remaining concerns and lead to an *increased rating of our work*.
>
> &nbsp;
>
> ## **Our Contributions**
>
> While several studies have attempted to mitigate the modality gap in CMFD [1, 2], to the best of our knowledge, our work is **the first to thoroughly analyze the impact of the modality gap on CMFD in relation to dimensional collapse and to propose the CIBA framework as a solution**.
> We acknowledge the reviewer's observation that the Information Bottleneck (IB) is widely used techniques in various applications.
> However, our contribution lies in leveraging these methods to specifically address the unique issue of dimensional collapse in CMFD, which we believe represents a significant and pioneering advancement in this domain.
>
> Furthermore, we have demonstrated **the effectiveness of the CIBA framework across various real-world datasets**, including large-scale datasets such as VGG-Sound (audio-video) and nuScenes (image-LiDAR).
> While the adopted methods, IB may appear straightforward and well-known, this does not diminish the significance of our analyses and demonstrations.
> Please note that our methodological contributions have been acknowledged and appreciated by reviewers "AFwU," "nuh2," and "I91G."
>
> > [1] Xue et al., The modality focusing hypothesis: Towards understanding crossmodal knowledge distillation, ICLR 2024
> [2] Sarkar1 et al., XKD: Cross-modal Knowledge Distillation with Domain Alignment for Video Representation Learning, AAAI 2024
>
> &nbsp;
>
> ## **Uniqueness of CIBA**
>
> In Section 3 and Fig.1, we theoretically and empirically demonstrate that transferring modality-general features from the teacher to the student is crucial for improving the quality of feature distillation. Based on **these unique insights**, we introduced the information bottleneck approach to approximate the modality-general features and distill them into sub-dimensional representations of the student’s features.
> Our approach fundamentally differs from prior CMKD methods, which primarily aim to **mimic the entire features of the teacher without considering potential dimensional collapse issues**. Furthermore, we have demonstrated the effectiveness of our CIBA framework across four different real-world multi-modal datasets: RAVDESS (Audio-Image), VGG-Sound (Audio-Video), MM-IMDB (Image-Text), and nuScenes (LiDAR-Camera).
>
>
> &nbsp;
>
> ## **Comparison to Output-level (Logit-level) Distillation Methods**
>
> Thank you for recommending interesting prior works from the knowledge distillation literature. We reviewed the details of each work and found them to be compelling, with a focus on output-level (logit-level) distillation. However, this focus seems to be slightly misaligned with our primary objective, which is **"feature-level distillation**".
> DML [1] utilizes multiple students and promotes their collaborative learning by minimizing the discrepancy in predictions across students, while DKD [2], DIST [3], and C2KD [4] propose methods to exploit the relationship between predictions and target class distributions. In particular, [1-3] focus on uni-modal distillation approaches, whereas C2KD [4] addresses the cross-modal distillation problem, which aligns with our motivation.
>
> We believe that **feature-level and output-level distillation strategies present distinct challenges**, necessitating thorough analyses to better understand their unique dynamics. While we recognize the importance of addressing dimensional collapse for achieving robust cross-modal feature distillation performance, **we did not explicitly analyze this issue at the output prediction stage**. Consequently, we are uncertain about how to directly compare our approach with existing methods in this context.
> However, we believe that combining feature-level and output-level distillation strategies could create synergies to further enhance the quality of CMKD. We leave this exploration for future work.
>
> > [1] Ying Zhang, Tao Xiang, Timothy M. Hospedales, and Huchuan Lu. Deep mutual learning. In CVPR, 2018.  \
> [2] Borui Zhao, Quan Cui, Renjie Song, Yiyu Qiu, and Jiajun Liang. Decoupled knowledge distillation. In CVPR, 2022.  \
> [3] Tao Huang, Shan You, Fei Wang, Chen Qian, and Chang Xu. Knowledge distillation from a stronger teacher. In NeurIPS, 2022.  \
> [4] Huo F, Xu W, Guo J, et al. C2KD: Bridging the Modality Gap for Cross-Modal Knowledge Distillation, Proceedings of the IEEE/CVF Conference on Computer Vision and Pattern Recognition. 2024: 16006-16015.

---

> ### Author Response · Authors · 2024-11-22
>
> ## **Relaxation of Orthogonality Assumption**
>
> As described in Section 3.6, the orthogonality assumption may not hold in real-world datasets since learned features often contain a mixture of modality-general information, modality-specific information, and sensor noise. Meanwhile, we have shown that **dimensional collapse occurs even in nonlinear real-world datasets**, as depicted in Fig.5(a). Furthermore, we have demonstrated that **addressing such issues significantly improves the quality of CMFD**, as evidenced by our extensive experimental results (Tab.1, 2, and 3).
>
> To further address the reviewer's concern, we conducted additional experiments where the orthogonality assumption was removed from the synthetic datasets discussed in Section 3.5. Specifically, we omitted the Gram-Schmidt process during synthetic data generation, meaning the data were generated following a unit normal Gaussian distribution and were **not fully orthogonal**.
> As shown in Fig.10 in Appendix E.6, The spectrum of singular values of the student features also decreases as the dimension of modality-general features decreases, although the distributions are less distinctive compared to those of orthogonal features (Fig.3). This reduction in the spectrum indicates the occurrence of dimensional collapse. Therefore, **our claims regarding modality-general information and dimensional collapse remains valid even in scenarios where the assumption of orthogonality is relaxed**.
>
> &nbsp;
>
> ## **Ablation on Hyperparameter $H$**
>
> As the reviewer pointed out, the bottleneck dimension parameter $H$ plays a crucial role in determining the quality of cross-modal feature distillation.
> We also elaborated that the optimal value of $H$ is proportional to the amount of modality-general information present in the teacher’s features (L427-431) and is also influenced by the method used to transfer this information (L410-413).
> In addition to the ablation results from RAVDESS presented in Fig.5(c), we also have showcased the impact of $H$ in LiDAR-Camera cross-modal feature distillation in Tab.2 by alternating the parameter $H$, where $H=4$ shows the best distillation performance.
> To further address the reviewer's concern, we provide **the ablation of $H$ on MM-IMDB and VGGSound dataset in the revised Appendix E.7**.
> Notably, except for extreme values of $H$, **the proposed method consistently outperforms the MSE method**, as shown in Fig.11 and 12.

---

> ### Comment · Reviewer_XgGs · 2024-11-26
> **Official Comment by Reviewer XgGs**
>
> Thank you for your response. I'd like to update the score accordingly.

---

> > ### Author Response · Authors · 2024-11-26
> >
> > Thank you for your response and for increasing your rating (from 3 to 5).
> >
> > We regret that your decision is marginally below acceptance. To facilitate a constructive discussion, we kindly request that you further elaborate on any remaining concerns that make you hesitant to rate our work with a clear acceptance. We believe this would greatly assist us in improving our work, and we are always ready to address any additional concerns you may have.
> >
> > We sincerely appreciate your careful consideration and the time and effort you have dedicated to this review again.

---

### Official Review · Reviewer_9LAr · 2024-11-07

**Soundness:** 2
**Presentation:** 3
**Contribution:** 2
**Rating:** 6
**Confidence:** 4

**Summary:**

In this paper, the author mainly propose to solve the problem of dimensional collapse caused by modality gap in cross-modal knowlegde distillation task. Firstly, the author demonstrates the impact of modality gap on cross-modal features theoretically and empirically. To combat with this issue, a Cross-modal Information Bottleneck Approximation (CIBA) framework is proposed that extracts modality-general features through a bottleneck structure, meanwhile aligning teacher and student features with an additional loss. Experiments on several datasets demonstrates the performance of CIBA.

**Strengths:**

1. The paper is well-written, which is quite easy to follow.
2. The author sufficiently demonstrates that modality gap can cause dimensional collapse, leading to suboptimal performance.

**Weaknesses:**

1. My main concern is about the generalizability of the method. As mentioned in problem statements and limitations, the theorem is established on linear extractor, which is inconsistent with practical applications where non-linear encoders are widely applied. Under such conditions, the proposed concept of modality-general and modality-specific parts could be vague since the better capacity of the encoders. I can truly understand the difficulty of theorical provement with non-linear extractors, while the direct application of CIBA seems to be accessible. Can you provide further results of CIBA compared to SOTAs with more powerful encoders on current datasets (RAVDESS and MM-IMDB) to prove the superiority?
2. In the method part, a bottleneck structure is utilized to capture mutual modality information. From my point of view, the dimension of the bottleneck feature may be a crucial parameter affecting the granularity of the extracted information. Performance seems to be fluctuant with chaning values of the param according to Fig.5(c). Can you provide more ablation on this param on more datasets? How do you choose the best bottleneck dimension?
3. The author mainly focus on the introduction and demonstration of modality gap's impact on dimensional collapse, while the introduction of method seems to be ordinary and unremarkable. Besides, since the information bottleneck structure was proposed by earlier research, and the proposed loss is a direct combination of generation loss and KL loss, the novelty of the paper is somehow limited.

**Questions:**

Please refer to the cons part. I will moderately raise my score if the authors can provide further experimental results and answer my questions.

---

> ### Author Response · Authors · 2024-11-22
>
> We appreciate the constructive comments from reviewer "9LAr", recognizing the strengths of our work as **writing quality** and **identifying the effect of the modality gap on the dimensional collapse**.
>
> We hope that our comments below adequately address your remaining concerns and lead to an *increased rating of our work*.
>
> &nbsp;
>
> ## **Robustness with Respect to Encoder Capacity**
>
> As discussed in Section 3.6, learned features in practical settings often contain a mixture of modality-general information, modality-specific information, and sensor noise.
> We acknowledge the reviewer's concern that the degree of such information mixing may vary depending on the capacity of the encoder.
> To rigorously address this concern and demonstrate the generalizability of our approach, we conducted additional experiments on the large-scale VGG-Sound dataset using backbones of varying sizes (ResNet-18 and ResNet-50).
> The results are presented in Section 5.4 and Appendix E.2 of the revised document.
> For your convenience, we provide a summary of the key findings in the table below.
>
> >**VGG-Sound**
> | Method | Modality | Val. | Test |
> |---|:---:|:---:|:---:|
> | Video (ResNet50 backbone) | V50 | 50.42 | 49.43 |
> | Audio (ResNet50 backbone) | A50 | 69.55 | 68.76 |
> | Video (ResNet18 backbone) | V18 | 42.11 | 41.53 |
> | Audio (ResNet18 backbone) | A18 | 68.86 | 69.08 |
> |||||
> | MSE         | V18 $\rightarrow$ A18 | 67.54 | 68.32 |
> | MSE + CIBA  | V18 $\rightarrow$ A18 | 70.11 | 70.39 |
> |||||
> | MSE         | V50 $\rightarrow$ A18 | 68.53 | 68.54 |
> | MSE + CIBA  | V50 $\rightarrow$ A18 | 70.21 | 70.71 |
> |||||
> | MSE         | A18 $\rightarrow$ V18 | 42.61 | 41.28 |
> | MSE + CIBA  | A18 $\rightarrow$ V18 | 43.59 | 42.55 |
> |||||
> | MSE         | A50 $\rightarrow$ V18 | 41.40 | 40.33 |
> | MSE + CIBA  | A50 $\rightarrow$ V18 | 43.44 | 42.95 |
>
> Experiments show that our approach consistently outperforms the vanilla distillation strategy (i.e., MSE loss).
> These findings confirm that **the CIBA framework retains its superiority regardless of the encoder's capacity**.
>
> &nbsp;
>
> ## **Our Contributions**
>
> While several studies have attempted to mitigate the modality gap in CMFD [1, 2], to the best of our knowledge, our work is the **first to thoroughly analyze the impact of the modality gap on CMFD in relation to dimensional collapse and to propose the CIBA framework as a solution**.
> We acknowledge the reviewer's observation that the Information Bottleneck (IB) and KL loss are widely used techniques in various applications.
> However, our contribution lies in leveraging these methods to specifically address **the unique issue of dimensional collapse in CMFD**, which we believe represents a significant and pioneering advancement in this domain.
> Furthermore, we have demonstrated **the effectiveness of the CIBA framework across various real-world datasets**, including large-scale datasets such as VGG-Sound (audio-video) and nuScenes (image-LiDAR).
> While the adopted methods, IB and KL loss, may appear straightforward and well-known, this does not diminish the significance of our analyses and demonstrations.
> Please note that our methodological contributions have been acknowledged and appreciated by reviewers "XgGs," "AFwU," "nuh2," and "i91g."
>
> > [1] Xue et al., The modality focusing hypothesis: Towards understanding crossmodal knowledge distillation, ICLR 2024
> [2] Sarkar1 et al., XKD: Cross-modal Knowledge Distillation with Domain Alignment for Video Representation Learning, AAAI 2024

---

> ### Author Response · Authors · 2024-11-22
>
> ## **Ablation on Hyperparameter $H$**
>
> As the reviewer pointed out, the bottleneck dimension parameter $H$ plays a crucial role in determining the quality of cross-modal feature distillation.
> We also elaborated that the optimal value of $H$ is proportional to the amount of modality-general information present in the teacher’s features (L427-431) and is also influenced by the method used to transfer this information (L410-413).
> In addition to the ablation results from RAVDESS presented in Fig.5(c), we also have showcased the impact of $H$ in LiDAR-Camera cross-modal feature distillation in Tab.2 by alternating the parameter $H$, where $H=4$ shows the best distillation performance.
> To further address the reviewer's concern, we provide the ablation of $H$ on the MM-IMDB and the additional VGGSound dataset in the Appendix E.7 of the revised document.
> Notably, except for extreme values of $H$, the proposed method consistently outperforms the MSE method, as shown in Fig.11 and 12 in the revised manuscript.
>
> &nbsp;
>
> ## **Computationally Efficient Surrogate for Estimating $H$**
>
> Instead of performing a grid search with training student models, we may use the bottleneck models as an efficient surrogate to estimate the optimal $H$ with relatively low computational cost, as described in Section 5.1.3.
> As an example, we first train bottleneck models with various $H$ values. Since the bottleneck model is significantly lighter than both the student and teacher models, it requires substantially fewer computational resources for training. We then analyze the singular value spectrum of the trained bottleneck features, as illustrated in Fig.5(b). Subsequently, we may select the minimum $H$ value at which the spectrum saturates. As demonstrated in Fig.5(c), this value consistently yields near best performance.
>
> In Appendix E.7, we extended the spectrum analyses presented in Fig.5 to the VGG-Sound dataset and observed similar results. Specifically, the spectrum (first row of Fig.12) converges around $H=16$. $H=16$ consistently achieves results close to the best (second row of Fig.12). Moreover, except for extreme values of $H$, the proposed method consistently outperforms the MSE method.

---

> ### Comment · Reviewer_9LAr · 2024-11-26
>
> Thank you for your response and additional experiments, my main concerns have been answered. I'd like to raise my score to 6.

---

> > ### Author Response · Authors · 2024-11-26
> >
> > Thank you for your positive feedback and for supporting our work by increasing your rating towards acceptance (from 5 to 6). We are glad to hear that your main concerns have been resolved. We sincerely appreciate your careful consideration and the time and effort you have dedicated to this review again.

---

### Author Response · Authors · 2024-11-22

We would like to express our sincere gratitude to all reviewers for their thoughtful and constructive comments on our manuscript. In this rebuttal, we have carefully addressed each comment raised by the reviewers. For ease of reference, we have organized our responses to individual comments under their respective points.

Additionally, in the revised version of the manuscript, all changes have been highlighted in **blue** to clearly indicate the modifications made in response to the reviewers’ suggestions. Thank you again for your time and effort in reviewing our work.

---

### Author Response · Authors · 2024-11-25
**Respectful Request for Rebuttal Feedback**

Dear All Reviewers,

We sincerely appreciate the time and effort that you have dedicated to reviewing our paper. Considering the significance of the discussion phase, we would like to ask if you could spare some time to review our rebuttal responses provided in the global and individual comments.

Please let us know if you have any areas needing clarification or follow-up questions. Your expertise and feedback are valuable to us, and we are fully prepared to handle any further questions and provide the necessary clarifications.

We sincerely hope that this discussion will help address the reviewers' remaining concerns and lead to an improved evaluation of our work.

Best regards, \
The authors.

---

### Author Response · Authors · 2024-12-03
**Final Comments from the Authors**

Dear all reviewers,\
\
We would like to extend our sincere gratitude for your constructive feedback and thoughtful engagement during the discussion period.

As the discussion period comes to a close, we would like to highlight that **no additional questions or concerns were raised following our rebuttal**. We hope this indicates that **our responses have sufficiently addressed your comments and resolved any outstanding issues**.

In this context, we kindly and respectfully ask you to **consider an upward adjustment to your evaluation score**.

Thank you once again for your time, effort, and valuable contributions.

\
Best regards,\
Authors

---

### Meta-Review · Area_Chair_9Ror · 2024-12-20

**Metareview:**

The submission investigates the issue of dimensional collapse in cross-modal knowledge distillation (CMKD) caused by the modality gap between teacher and student models. The authors propose the Cross-modal Information Bottleneck Approximation (CIBA) framework to mitigate this issue by disentangling modality-general and modality-specific features. While the paper is well-written and supported by extensive experiments, the novelty and contributions are deemed insufficiently compelling for acceptance at this stage.

Related work in modality gap analysis and logit-level distillation (e.g., C2KD) is insufficiently addressed, and comparisons to state-of-the-art techniques are either missing or not directly relevant. Theoretical analyses are based on linear feature extractors, which may not generalize to the non-linear encoders used in practical applications.  Several reviewers (e.g., XgGs, i91g, AFwU) retained their scores near the acceptance threshold, citing unresolved concerns about novelty, experimental robustness, and comparisons to advanced baselines.

**Additional Comments On Reviewer Discussion:**

Multiple reviewers (9LAr, XgGs, i91g) highlighted that the primary contributions rely on established methods such as the Information Bottleneck (IB) framework and a combination of generation loss and KL loss.

---

### Decision · Program_Chairs · 2025-01-22

Reject